# Routine single particle CryoEM sample and grid characterization by tomography

Alex J Noble[1], Venkata P Dandey[1†], Hui Wei[1†], Julia Brasch[1,2], Jillian Chase[3,4], Priyamvada Acharya[1,5], Yong Zi Tan[1,2], Zhening Zhang[1], Laura Y Kim[1], Giovanna Scapin[1,6], Micah Rapp[1,2], Edward T Eng[1], William J Rice[1], Anchi Cheng[1], Carl J Negro[1], Lawrence Shapiro[2], Peter D Kwong[5], David Jeruzalmi[3,4,7,8], Amedee des Georges[3,4,8,9], Clinton S Potter[1,2], Bridget Carragher[1,2]*

[1]National Resource for Automated Molecular Microscopy, Simons Electron Microscopy Center, New York Structural Biology Center, New York, United States; [2]Department of Biochemistry and Molecular Biophysics, Columbia University, New York, United States; [3]Department of Chemistry and Biochemistry, City College of New York, New York, United States; [4]Program in Biochemistry, The Graduate Center of the City University of New York, New York, United States; [5]Vaccine Research Center, National Institute of Allergy and Infectious Diseases, National Institutes of Health, Maryland, United States; [6]Department of Structural Chemistry and Chemical Biotechnology, Merck & Co., Inc, New Jersey, United States; [7]Program in Biology, The Graduate Center of the City University of New York, New York, United States; [8]Program in Chemistry, The Graduate Center of the City University of New York, New York, United States; [9]Advanced Science Research Center, The Graduate Center of the City University of New York, New York, United States

*For correspondence: bcarr@nysbc.org

†These authors contributed equally to this work

Competing interests: The authors declare that no competing interests exist.

**Abstract** Single particle cryo-electron microscopy (cryoEM) is often performed under the assumption that particles are not adsorbed to the air-water interfaces and in thin, vitreous ice. In this study, we performed fiducial-less tomography on over 50 different cryoEM grid/sample preparations to determine the particle distribution within the ice and the overall geometry of the ice in grid holes. Surprisingly, by studying particles in holes in 3D from over 1000 tomograms, we have determined that the vast majority of particles (approximately 90%) are adsorbed to an air-water interface. The implications of this observation are wide-ranging, with potential ramifications regarding protein denaturation, conformational change, and preferred orientation. We also show that fiducial-less cryo-electron tomography on single particle grids may be used to determine ice thickness, optimal single particle collection areas and strategies, particle heterogeneity, and de novo models for template picking and single particle alignment.
DOI: https://doi.org/10.7554/eLife.34257.001

## Introduction

For decades, single particle cryo-electron microscopy (cryoEM) grids have commonly been imaged and processed under the assumption that most particles imaged were not adsorbed to the air-water interfaces and were in a single layer as they were plunge-frozen. An ideal grid and sample for single particle collection would have the majority of areas in holes maximally occupied by non-adsorbed, non-interacting particles 10 nm or farther from the air-water interfaces, particles oriented randomly, vitreous ice thin enough to contain the particles plus about 20 nm of additional space, where none of the particles overlap in the beam direction, and where the beam direction is normal to the areas

of interest (*Figure 1*). Collection in such ideal areas of a grid would then be the most efficient use of resources and would result in the highest resolution structure possible for a given number of particles, collection hardware, and collection parameters.

In practice, during single particle grid preparation and data collection there are many issues that contribute to preventing a sample from following this ideal behavior. As depicted in *Figure 2*, numerous combinations of air-water interface, particle, and ice behavior are possible for each hole and for regions within each hole, without taking into account surface ice contamination. Each air-water interface might be: (i) free from sample solution constituents (*Figure 2*, A1), (ii) covered with a layer of primary, secondary, and/or tertiary protein structures (either isolated or forming protein networks) from denatured particles (A2), or (iii) covered with one or more layers of surfactants if present during preparation (A3). It is difficult to distinguish between air-water interfaces that are clean, covered in primary protein structures, or covered in surfactants as they are likely indistinguishable by cryoEM or cryo-electron tomography (cryoET) analysis without a sample-free control for comparison (cryoET may be able to resolve lipid layers at the air-water interface if high tilt angles are collected [*Vos et al., 2008*]). Bulk particle behavior in regions of holes might include any combination of: (i) non-adsorbed particles without preferred orientation (B1), (ii) particles at an air-water interface without preferred orientation (B2), (iii) particles at an air-water interface with N-preferred orientations (B3), (iv) partially denatured particles at an air-water interface with M-preferred orientations (B4), and/or (v) significantly denatured particles at an air-water interface (B5). Protein degradation in A2 might be considered to be a continuation of the denaturation in B4 and B5. Interactions between neighboring particles at the air-water interface might induce different preferred orientations in B3 and B4, particularly at high concentrations. Ice behavior at the air-water interfaces of each hole might be characterized by any two combinations of: convex ice (C1), flat ice (C2), concave ice where the center is thicker than the particle's minor axis (C3), and/or concave ice where the center is thinner than the particle's minor axis (C4). In the case of a convex air-water interface, the particle's minor axis might be larger than the ice thickness at the edge of the hole.

The most common technique for preparing cryoEM grids, pioneered in the labs of Robert Glaeser (*Jaffe and Glaeser, 1984*; *Taylor and Glaeser, 1974*; *Taylor and Glaeser, 1976*) and Jacques Dubochet (*Adrian et al., 1984*; *Dubochet et al., 1985*; *Dubochet et al., 1982*), involves applying about 3 microliters of purified protein in solution onto a metal grid covered by a holey substrate that has been glow-

discharged to make hydrophilic, blotting the grid with filter paper, and plunge-freezing the grid with remaining sample into a cryogen to form vitreous ice. Incubation times before and after blotting are on the order of seconds, allowing for the possibility of protein adsorption to the air-water interface due to Brownian motion. Concerns regarding deleterious air-water interface interactions with proteins have been often discussed in the literature. For instance, Jacques *Dubochet et al., 1988* observed issues with regards to air-water interface and particle orientation for a small number of samples. In a recent review by Robert Glaeser (*Glaeser and Han, 2017*), evidence (*Trurnit, 1960*) using Langmuir-Blodgett (LB) troughs (*Langmuir, 1917*) was used to propose that upon contact with a clean air-water interface, proteins in solution will denature, forming an insoluble, denatured protein film. This film reduces the surface tension at the air-water interface and might act as a barrier between the remaining particles in solution and the air. Particles in solution might then adsorb to the denatured layer of protein depending on the local particle affinity with the interface, thus creating an ensemble of preferred orientations. Estimates for the amount of time a particle with a mass of 100 kDa to 1 MDa in solution might take to first reach the air-water interface (bulk diffusion) range from 1 ms to 0.1 s (*Naydenova and Russo, 2017*; *Taylor and Glaeser, 2008*).

More recent literature, using LB troughs, substantiates that 10–1000 mL volumes of various proteins (commonly 10–1000 kDa and at ≤1 mg/mL) in buffer commonly adsorb to the air-water interface and form <10 nm thick (*Gunning et al., 1996*; *Vliet et al., 2002*) denatured viscoelastic protein network films (*Birdi, 1972*; *Damodaran and Song, 1988*; *de Jongh et al., 2004*; *Dickinson et al., 1988*; *Graham and Phillips, 1979*; *Yano, 2012*). The time it takes for adsorption to begin due to bulk diffusion may be on the order of 0.1 to 1 ms, depending on the protein (*Kudryashova et al., 2005*). For a protein that denatures at the air-water interface (surface diffusion), the surface diffusion time might be on the order of tens of milliseconds (*Kudryashova et al., 2005*), depending on factors including protein and concentration, surface hydrophobicity, amount of disordered structure, secondary structure, concentration of intramolecular disulfide bonds, buffer, and temperature.

Higher bulk protein concentrations have been shown to increase the protein network thickness (*Meinders et al., 2001*). When several proteins and/or surfactants in solution are exposed to a clean air-water interface, competitive and/or sequential adsorption may occur (*Ganzevles et al., 2008*; *Le Floch-Fouéré et al., 2010*; *Stanimirova et al., 2014*). It has been shown using atomic force microscopy (AFM) imaging of LB protein films that these protein network films may not completely denature down to individual amino acids: adding surfactants to protein solutions in which a protein network film has already formed at the air-water interface will displace the protein layer (desorption [*MacRitchie, 1998*]) (*Gunning and Morris, 2018*; *Mackie et al., 1999*; *Wilde et al., 2004*) and the resulting protein network segments might partially re-fold in solution (*Gunning and Morris, 2018*; *Mackie et al., 1999*; *Morris and Gunning, 2008*). Time-resolved AFM surfactant-protein displacement experiments for a specific protein, β-lactoglobulin, and different surfactants, Tween 20 and Tween 60, show that displacement of the protein network film by the surfactants occurs at equivalent surface pressures and results in non-uniform surfactant domain growth, implying that the protein network is not uniform (*Gunning et al., 2004*). Different surfactant displacement behavior and patterns are observed while varying only the proteins, where the degree of protein network displacement isotropy by surfactant decreases for more ordered, globular proteins (*Mackie et al., 1999*). Non-uniformity of the protein network has also been seen by 3D AFM imaging of β-lactoglobulin LB-protein network films placed on mica (*Gunning et al., 1996*; *Morris and Gunning, 2008*). Similar

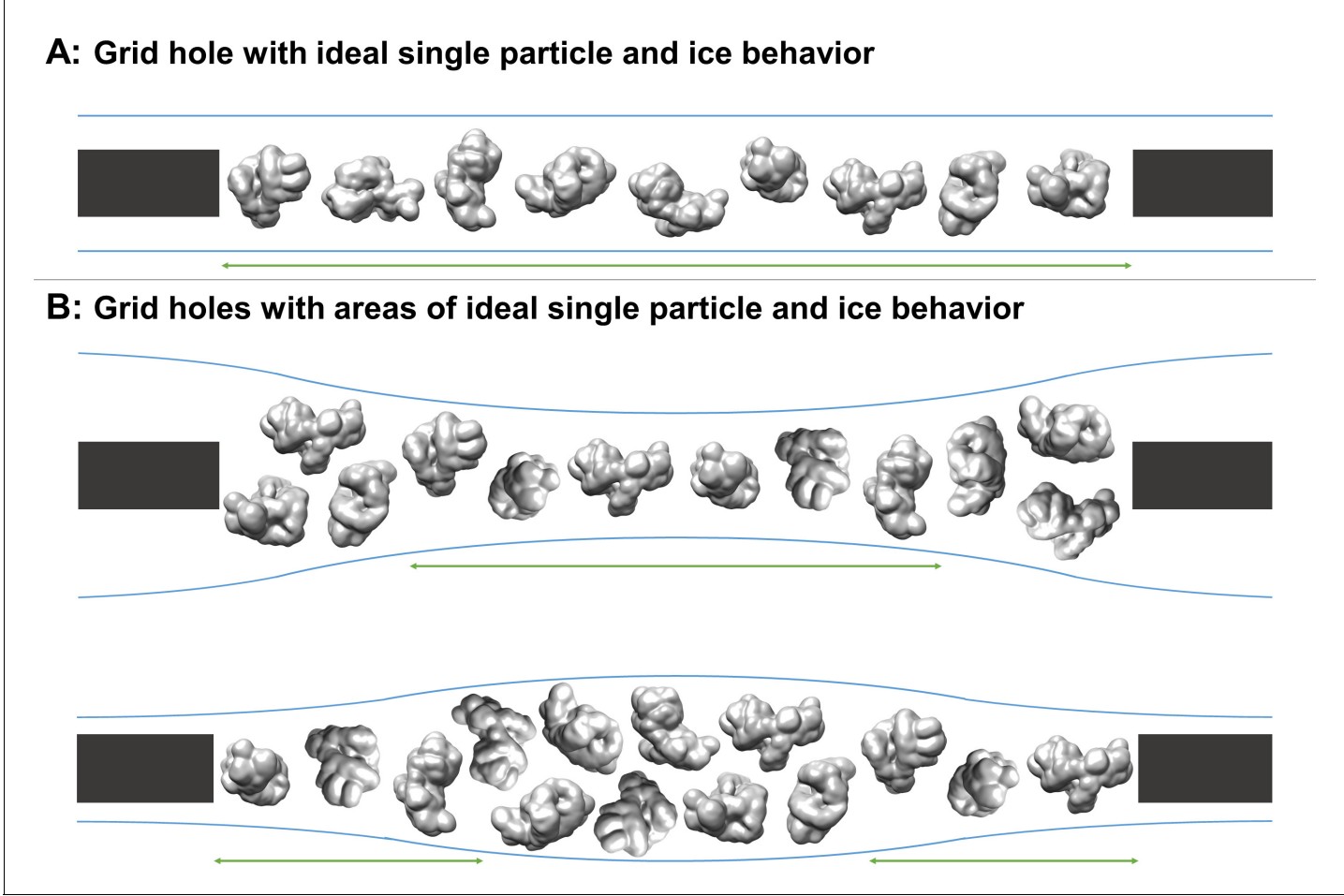

**A: Grid hole with ideal single particle and ice behavior**

**B: Grid holes with areas of ideal single particle and ice behavior**

**Figure 1.** Schematic diagrams of grid hole cross-sections containing regions of ideal particle and ice behavior for single particle cryoEM collection. (A) A grid hole where all regions of particles and ice exhibit ideal behavior. (B) Grid holes where there are areas that exhibit ideal particle and ice behavior. Green arrows indicate areas with ideal particle and ice behavior. The generic particle shown is a low-pass filtered holoenzyme, EMDB-6803 (*Yin et al., 2017*). The particles were rendered with UCSF Chimera (*Pettersen et al., 2004*).
DOI: https://doi.org/10.7554/eLife.34257.002

## A: Potential air-water interface composition

1) Clean

2) Primary, secondary, tertiary protein structures/networks from denaturation

3) Surfactants (if present)

## B: Potential bulk particle behavior at/near an air-water interface*

1) Non-adsorbed particles (no preferred orientation)

2) Particles at air-water interface (no preferred orientation)

3) Particles at air-water interface, no denaturation (N-preferred orientations)

4) Particles at air-water interface, partial denaturation (M-preferred orientations)

5) Particles at air-water interface, significant denaturation

\* Particles might also aggregate.

## C: Potential ice thickness variations in holes†

1) Convex

2) Flat

3) Concave (center is thicker than particle's minor axis)

4) Concave (center is thinner than particle's minor axis)

† Apposed ice curvatures are not necessarily equivalent.

**Figure 2.** Depictions of potential ice and particle behavior in cryoEM grid holes, based on *Figure 6* from (*Taylor and Glaeser, 2008*). A region of a hole may be described by a combination of one option from (**A**) for each air-water interface and one or more options from (**B**). An entire hole may be described by a set of regions and one or more options from (**C**). (**A**) Each air-water interface might be described by either (1), (2), or (3). Note that cryoET might only be able to resolve tertiary and secondary protein structures/network elements at the air-water interface. (**B**) Particle behavior between air-water interfaces and at each interface might be composed of any combination of (1) through (5), with or without aggregation. B3 is different from B4 if, for example, a particle prone to denaturation is frozen before or after denaturation has begun, thus potentially changing the set of preferred orientations. At high enough concentrations additional preferred orientations might become available in B3 and B4 due to neighboring protein-protein interactions. (**C**) Ice thickness variations through a central cross-section of hole may be described by one option for one air-water interface and one option for the apposed interface. Note that in C1 the particle's minor axis may be larger than the ice thickness. In both C1 and C4,

*Figure 2 continued on next page*

*Figure 2 continued*

the particle may still reside in areas thinner than its minor axis if the particle is compressible. Phenomenon such as bulging or doming (*Brilot et al., 2012*) may be represented as a combination of C1-4.

DOI: https://doi.org/10.7554/eLife.34257.003

experiments using LB troughs have also shown that proteins with β-sheets partially unfold, with the hydrophobic β-sheets remaining in-tact at the air-water interface and with potentially one or more layers of unstructured, but connected, hydrophilic amino acid strands just below the air-water interface (*Yano et al., 2009*). This potential for β-sheets to survive bulk protein denaturation is likely due to β-sheets commonly consisting of alternating hydrophobic and hydrophilic (polar or charged) side-chains (*Zhang et al., 1993*), with the hydrophobic sidechains orienting towards the air. Intermolecular β-sheets may also bind together, strengthening the protein network (*Martin et al., 2005*; *Renault et al., 2002*). Moreover, the number of random coils, α-helices, and β-sheets for a protein in bulk solution might each increase or decrease when introduced to a hydrophobic environment (*Reddy and Nagara, 1989*; *Zangi et al., 2002*), including the air-water interface (*Martin et al., 2003*; *Yano, 2012*), implying that protein conformation when adsorbed to the air-water interface could be different than when in solution (*Lad et al., 2006*; *Vance et al., 2013*; *Yano, 2012*). Measurements of shear stress and compressibility of protein network films versus the internal cohesion of the constituent protein show a correlation: the more stable a protein in bulk solution, the more robust the resulting protein network film at the air-water interface (*Martin et al., 2005*). At high enough surface concentrations and depending on surface charge distribution, neighboring globular proteins might interact to induce additional preferred orientations as has been shown in surface-protein studies (*Billsten et al., 1995*; *Rabe et al., 2011*; *Tie et al., 2003*). Such nearest neighbor protein-protein interactions may in turn decrease protein affinity to the interface and increase desorption. Similar effects might occur at protein-air-water interfaces.

Given the length of incubation time commonly permitted before plunging a grid for cryoEM analysis, the cross-disciplinary research discussed above suggests that some particles in a thin film on a cryoEM grid will form a viscoelastic protein network film at the air-water interface. The composition and surface profile of the resulting protein network film will vary depending on the structural integrity of the bulk protein and the bulk protein concentration. Bulk protein affinity to the protein network film will then vary depending on the local affinity between the film and the proteins. To better understand the range of particle behaviors with respect to the air-water interfaces in cryoEM grid holes, a representative ensemble of grid and sample preparations needs to be studied in three dimensions.

One method of studying single particle cryoEM grids is using cryoET. CryoET is typically practiced by adding gold fiducials to the sample preparation for tilt-series alignment, which requires additional optimization steps and might not be representative of the same sample prepared without gold fiducials. To avoid the issues imposed by gold fiducials, we have employed the fiducial-less tilt-series alignment method of Appion-Protomo (*Noble and Stagg, 2015*), allowing for cryoET analysis of all single particle cryoEM grids we have attempted. We used this fiducial-less cryoET method to investigate over 50 single particle cryoEM samples sourced from dozens of users and using grids prepared using either conventional grid preparation techniques or the new Spotiton (*Jain et al., 2012*) method. Our aim was to determine the locations of particles within the vitreous ice and the overall geometry of the ice in grid holes (related to the possible combinations in *Figure 2*).

We have also found that the usefulness of performing cryoET on a single particle cryoEM grid extends beyond the goal of understanding the arrangements of particles in the ice. CryoET allows for the determination of optimal collection locations and strategies, single particle post-processing recommendations, understanding particle structural heterogeneity, understanding pathological particles, and de novo model building. We contend that cryoET should be routinely performed on single particle cryoEM grids in order to fully understand the nature of the sample on the grid and to assist with the entire single particle collection and processing workflow. We have made available a standalone Docker version of the Appion-Protomo fiducial-less tilt-series alignment suite used in these investigations at https://github.com/nysbc/appion-protomo.

**Table 1.** Ice thickness measurements, number of particle layers, preferred orientation estimation, and distance of particle layers from the air-water interface as determined by cryoET of single particle cryoEM grids for 46 grid preparations of different samples.

The table is ordered in approximate order of increasing particle mass. Several particles are un-named as they are yet to be published. Sample concentration in solution is included with the sample name if known. Distance measurements are measured with an accuracy of a few nanometers due to binning of the tomograms by a factor of 4 and estimation of air-water interface locations using either contamination or particle layers. Grid types include carbon and gold holey grids and lacey and holey nanowire grids, plunged using conventional methods or with Spotiton. Edge measurements are made ~100 nm away from hole edges. '–' indicates that these values were not measurable. Samples highlighted with blue contain regions of ice with near-ideal conditions (<100 nm ice, no overlapping particles, little or no preferred orientation). Samples highlighted with green contain regions of ice with ideal conditions (non-ideal plus no particle-air-water interface interactions). Incubation time for the samples on the grid before plunging is on the order of 1 s or longer.

| Sample # | Sample name | Grid type | Ice thickness (center, edge, substrate) in nm ± a few nm | | | # of Layers (center, edge, substrate) | | | Apparent preferred orientation in layer? | Min. particle/layer distance from air-water interface (nm ± a few nm) |
|---|---|---|---|---|---|---|---|---|---|---|
| 1* | 32 kDa Kinase | Carbon Spotiton | 65 | 45 | – | 0 | 0 | 0 | Unknown | <5 |
| 2 | 32 kDa Kinase | Gold Spotiton | 30 | – | – | 0 | – | – | Unknown | <5 |
| 3 | Insulin Receptor | Gold Spotiton | 55 | – | – | 1–2 | – | – | No | 5 |
| 4*† | Hemagglutinin | Carbon Spotiton | 25–95 | 100–210 | – | 0 or 2 | 2 | – | Some | 5 |
| 5* | HIV-1 Trimer Complex 1 | Carbon Spotiton | 75–210 | – | – | 2 | – | – | Yes | 5–10 |
| 6* | HIV-1 Trimer Complex 1 | Gold Spotiton | 20 | – | – | 1 | – | – | Some | 5 |
| 7* | HIV-1 Trimer Complex 2 | Carbon Spotiton | 190 | 265 | – | 2 | 2 | 2 | Yes | 5 |
| 8 | 147 kDa Kinase | Gold Spotiton | 15 | – | – | 1 | – | – | Unknown | <5 |
| 9 | 150 kDa Protein | Holey Carbon Spotiton | 35 | 70 | – | 2 | 2 | 2 | Some | <5 |
| 10* | Stick-like Protein 1‡ | Carbon Spotiton | 80 | – | – | 1 | – | – | No | <5 |
| 11 | Stick-like Protein 2 (150 kDa)‡ | Carbon CFlat | 100 | 100 | – | 1 | 1 | – | Unknown | 5 |
| 12* | Stick-like Protein 2‡ | Gold Spotiton | 135–190 | – | – | 1 | – | – | Some | 5 |
| 13* | Neural Receptor‡ | Carbon Spotiton | 60–90 | – | – | 1 | – | – | Yes | 5 |
| 14* | Neural Receptor‡ | Carbon Spotiton | 80–90 | 100–140 | 135 | 1 | 1 | 1 | Yes | 5 |
| 15 | 200 kDa Protein | CFlat Carbon + Gold mesh | 40–60 | 95 | 110 | 1 | 1 | 2 | No | 5 |
| 16 | Small, Popular Protein | Carbon Spotiton | 30 | 70 | – | 1 | 2 | 2 | No | 5 |
| 17* | Glycoprotein with Bound Lipids (deglycosylated) | Carbon Spotiton | 15 | 90 | 130 | 1 | 2 | 2 | Yes | <5 |
| 18 | Glycoprotein with Bound Lipids (deglycosylated)‡ | Gold Spotiton | 155 | – | – | 2 | – | – | Some | <5 |
| 19* | Lipo-protein | Holey Carbon | 0–95 | 85–100 | – | Uniformly distributed in ice | | | Unknown | 5 |
| 20* | GPCR | Carbon Spotiton | 25 | – | – | 1 | 2 | – | No | 5 |
| 21*† | Rabbit Muscle Aldolase (1 mg/mL) | Gold Spotiton | 15 | 50 | – | 1 | 2 | – | No | <5 |
| 22*† | Rabbit Muscle Aldolase (6 mg/mL) | Carbon Spotiton | 60–110 | 75–130 | 85 | 2 | 2 | 2 | Some | 5 |
| 23 | Un-named Protein | Holey Carbon | 35 | – | 60 | 1 | – | 2 | Yes | 5 |
| 25* | Protein in Nanodisc (0.58 mg/mL) | Gold Spotiton | 30 | 65 | – | 1–2 | 2 | – | No | 5–10 |
| 26 | IDE | Carbon Spotiton | 25 | 60 | 95 | 1 | 2 | 2 | Unknown | 5 |

*Table 1 continued on next page*

*Table 1 continued*

| Sample # | Sample name | Grid type | Ice thickness (center, edge, substrate) in nm ± a few nm | | | # of Layers (center, edge, substrate) | | | Apparent preferred orientation in layer? | Min. particle/ layer distance from air-water interface (nm ± a few nm) |
|---|---|---|---|---|---|---|---|---|---|---|
| 27* | IDE | Gold Spotiton | 40 | – | – | 1 | – | – | No | 5–10 |
| 28 | Small, Helical Protein | Gold Spotiton | 50 | 75 | – | 1 | 2 | – | Some | 5 |
| 29 | 300 kDa Protein | Carbon Spotiton | 30 | 100 | – | 1 | 2 | 2 | No | 5 |
| 30*† | GDH | Holey Carbon | 30 | 85 | 100 | 1 | 1 | 3 | Some | 5 |
| 31*† | GDH | Holey Carbon | 60 | 120 | 140 | 1 | 2 | 3 | Yes | 5 |
| 32*† | GDH (2.5 mg/mL)+0.001% DDM | Carbon Spotiton | 50 | 180 | 190 | 1 | 2 | – | Yes | <5 |
| 33*† | DnaB Helicase-helicase Loader | Gold Quantifoil | 50–55 | 80–100 | – | 1 | 2 | – | No | 5 |
| 34*† | Apoferritin | Gold Spotiton | 25–30 | – | – | 1 | – | – | No | 5 |
| 35*† | Apoferritin | Gold Spotiton | 25 | – | – | 1 | – | – | No | 5 |
| 36*† | Apoferritin | Holey Carbon Spotiton | 30 | 125 | 135 | 1 | 2 | 2 | No | 5 |
| 37*† | Apoferritin (1.25 mg/mL) | Holey Carbon Spotiton | 30–50 | 100 | 105 | 1 | 2 | 2 | No | 5 |
| 38*† | Apoferritin (0.5 mg/mL) | Holey Gold Spotiton | 25–30 | 55 | – | 1 | 2 | – | No | <5 |
| 39*† | Apoferritin with 0.5 mM TCEP | Carbon Spotiton | 40–90 | 145–175 | – | 1–2 | 2 | 1 | No | 5 |
| 40 | Protein with Carbon Over Holes | Carbon Quantifoil | 110 | 70–100 | – | 1 | 1 | – | Some | 5–10 |
| 41 | Protein and DNA Strands with Carbon Over Holes | Carbon Quantifoil | 60 | – | – | 1 | – | – | Some | 5–10 |
| 42*† | T20S Proteasome | Holey Carbon | 35 | 115 | 120 | 1 | 2 | 3 | Some | <5 |
| 43*† | T20S Proteasome | Holey Carbon | 125 | 140–160 | 150 | 2 | 2 | 2 | Some | 5 |
| 44*† | T20S Proteasome | Gold Quantifoil | 50–75 | – | – | 1 | – | – | Some | 5 |
| 45*† | Mtb 20S Proteasome | Carbon Spotiton | 35 | 80 | 115 | 0 | 1 | 1 | No | 5–10 |
| 46 | Protein on Streptavidin | Holey Carbon | 20–100 | 80–120 | – | 0–2 | 1–2 | – | No | 10 |

*A video is included for this sample.

†A dataset is deposited for this sample.

‡Intentionally thick ice.

DOI: https://doi.org/10.7554/eLife.34257.004

## Results and discussion

The fiducial-less tomography pipeline at the New York Structural Biology Center (NYSBC) consisting of Leginon (*Suloway et al., 2005*; *Suloway et al., 2009*) or SerialEM (*Mastronarde, 2003*) for tilt-series collection and Appion-Protomo (*Noble and Stagg, 2015*; *Winkler and Taylor, 2006*) for tilt-series alignment allows for the routine study of grids and samples prepared for single particle cryoEM in three dimensions. The resulting analysis sheds light on long standing questions regarding how single particle samples prepared using traditional methods (manual, Vitrobot, and CP3 plunging), or with new automated plunging with Spotiton (*Jain et al., 2012*), behave with respect to the air-water interfaces. In the following sections, we report and discuss how tomography collection areas were determined and analyzed, the observation that the vast majority of particles are local to the air-water interfaces and the implications with regards to potential denaturation, the prevalence of overlapping particles in the direction orthogonal to the grid, the observation that most cryoEM imaging areas and particles are tilted several degrees with respect to the electron beam, the value of cryoET to determine optimal collection locations and strategies, the benefits of using cryoET to

**Table 2.** Apparent air-water interface, particle, and ice behavior of the same samples in **Table 1** using the descriptions in **Figure 1**. Tilt-series were aligned and reconstructed using the same workflow and thus are oriented in the same direction. However, the direction relative to the sample application is not known. The bottom air-water interface corresponds to lower z-slice values, and the top to higher z-slice values as rendered in 3dmod from the IMOD package (**Kremer et al., 1996**). 'A' means that the air-water interface is apparently clean and cannot be visually differentiated between A1, A2 (primary structure), or A3. Percentages in parentheses are particle layer saturation estimates. Reported angles are the angles (absolute value) between the particle layer's normal and the electron beam direction, measured using 'Slicer' in 3dmod. It is often difficult to distinguish between flat and curved ice at the air-water interfaces (e.g. **Figure 2**, 'C1 or C2' or 'C2 or C3') because most fields of view do not span entire holes. '‡' indicates that the top layer of objects is the same layer as the bottom layer. '–' indicates that these values were not measurable.

| Sample # | Sample name | Air-water interface, particle behavior, and layer/ice angle (bottom, center) | Air-water interface, particle behavior, and layer/ice angle (bottom, edge) | Ice behavior (bottom) | Air-water interface, particle behavior, and layer/ice angle (top, center) | Air-water interface, particle behavior, and layer/ice angle (top, edge) | Ice behavior (top) | Notes |
|---|---|---|---|---|---|---|---|---|
| 1* | 32 kDa Kinase | A, B1 or B2 or B3 (50%), 8° | A, B1 or B2 or B3 (50%), 10° | C2 | A, B1 or B2 or B3‡ (50%), 8° | A, B1 or B2 or B3‡ (50%), 10° | C2 | Particles aggregate into clouds. |
| 2 | 32 kDa Kinase | A, B1 or B2 or B3 (50%), 4–8° | – | C1 or C2 | A, B1 or B2 or B3‡ (50%), 4–8° | – | C1 or C2 | Gold beads are glow discharge contamination. |
| 3 | Insulin Receptor | A, B1 or B2 or B3 (100%), 3–5° | – | C2 or C3 | A, B1 or B2 or B3‡ (100%), 3–5° | – | C2 or C3 | Gold beads are glow discharge contamination. |
| 4*† | Hemagglutinin | A2, No particles, 3–7° | A, B3 (40%), 5° or A, B3 (40%), 3° | C3 or C4 | A2‡, No particles, 3–7° or A, B3 (50%), 7° | A, B3 (50%), 5–7° | C3 or C4 | Where very thin ice in the center of holes excludes particles, protein fragments remain. |
| 5* | HIV-1 Trimer Complex 1 | A2, B1, B3 (30%), 1–5° | – | C1, C2, or C3 | A2, B1, B3 (30%), 1–5° | – | C1, C2, or C3 | Trimer domains and/or unbound receptors are adsorbed to air-water interfaces. |
| 6* | HIV-1 Trimer Complex 1 | A2, B3 (80%), 6° | – | C2 | A2, B3‡ (80%), 6° | – | C2 | Trimer domains and/or unbound receptors are adsorbed to air-water interfaces. |
| 7* | HIV-1 Trimer Complex 2 | A, B2 or B3 (50%), 1° | A, B2 or B3 (50%), 3° | C1 or C2 | A, B2 or B3 (70%), 1° | A, B2 or B3 (70%), 3° | C1 or C2 | |
| 8 | 147 kDa Kinase | A, B2 or B3 (50%), 0° | – | C2 or C3 | A, B2 or B3‡ (50%), 0° | – | C2 or C3 | Gold beads are glow discharge contamination. |
| 9 | 150 kDa Protein | A, B2 or B3 (60%), 7–10° | A, B2 or B3 (60%), 8° | C2 or C3 | A, B2 or B3‡ (60%), 7° | A, B2 or B3 (40%), 9° | C2 or C3 | |
| 10* | Stick-like Protein 1 | A and A2, B4 and B5 (1%), 10° | – | C2 | A2, B4 and B5 (50%), 10° | – | C2 | |
| 11 | Stick-like Protein 2 (150 kDa) | A2, B3 and B4 and B5 (70%), 7° | A2, B3 and B4 and B5 (70%), 7° | – | A2, B3 and B4 and B5‡ (70%), 7° | A2, B3 and B4 and B5‡ (70%), 7° | – | Determinations are not accurate due to over focusing and minimal tilt angles. |
| 12* | Stick-like Protein 2 | A2, B3 (80%), 0° | – | C2 or C3 | A2, B3 (1%), 0° | – | C2 or C3 | Note 1. Note 2. |
| 13* | Neural Receptor | A2, B3 (80%), 3–10° | – | C2 or C3 | A2, No particles, 3–10° | – | C2 or C3 | Note 1. Note 2. |
| 14* | Neural Receptor | – | A2, No particles, 2–7° or A2, B3 (70%), 5° | C3 | – | A2, B3 (70%), 7° or A2, No particles, 7° | C3 | Note 1. Note 2. Two tomograms have one orientation, one has the opposite. |
| 15 | 200 kDa Protein | A, B2 or B3 (60%), 2° | A, B2 or B3 (50%), 4° | C3 | No particles or A, B2 or B3‡ (60%), 2° | A, No particles, 11° | C3 | |
| 16 | Small, Popular Protein | A, B2 or B3 (90%), 6° | A, B2 or B3 (90%), 9° | C2 | A, B2 or B3‡ (90%), 6° | A, B2 or B3 (90%), 1° | C3 | |

*Table 2 continued on next page*

Table 2 continued

| Sample # | Sample name | Air-water interface, particle behavior, and layer/ice angle (bottom, center) | Air-water interface, particle behavior, and layer/ice angle (bottom, edge) | Ice behavior (bottom) | Air-water interface, particle behavior, and layer/ice angle (top, center) | Air-water interface, particle behavior, and layer/ice angle (top, edge) | Ice behavior (top) | Notes |
|---|---|---|---|---|---|---|---|---|
| 17* | Glycoprotein with Bound Lipids (deglycosylated) | A, B3 (70%), 4° | A, B3 (80%), 10° | C3 | A, B3$^{\ddagger}$ (70%), 4° | A, B3 (80%), 11° | C3 | Lipid membrane dissociates from protein in center. |
| 18 | Glycoprotein with Bound Lipids (glycosylated) | A, B3 (50%), 10° | – | C2 or C3 | A, B3 (60%), 4° | – | C2 or C3 | |
| 19* | Lipo-protein | No particles or A, B2, 3° | A, B3, 11° | C3, C4 | No particles or A, B2$^{\ddagger}$, 5° | A, B3, 11° | C3, C4 | Particles are uniformly distributed in the ice. |
| 20* | GPCR | A, B2 or B3 (70%), 3° | A, B2 or B3 (60%), – | C3 | A, B2 or B3$^{\ddagger}$ (70%), 3° | A, B2 or B3 (60%), – | C3 | |
| 21*† | Rabbit Muscle Aldolase (1 mg/mL) | A, B2 or B3 (90%), 3–9° | A, B2 or B3 (80%), 6° | C3 | A, B2 or B3$^{\ddagger}$ (90%), 3–9° | A, B2 or B3 (80%), 10° | C3 | |
| 22*† | Rabbit Muscle Aldolase (6 mg/mL) | A, B1, B2 or B3 (90%), 5° | A, B1, B2 or B3 (90%), 5° | C2 or C3 | A, B1, B2 or B3 (90%), 5° | A, B1, B2 or B3 (90%), 5° | C2 or C3 | |
| 23 | Un-named Protein | A, B3 (40%), 0–3° | – | C2 or C3 | A, B3$^{\ddagger}$ (40%), 0–3° | – | C2 or C3 | |
| 24 | Un-named Protein | A, B3 (80%), 2° | A, B3 (60%), 4–6° | C3 | A, B3$^{\ddagger}$ (80%), 2° | A, B3 (60%), 4–9° | C3 | |
| 25* | Protein in Nanodisc (0.58 mg/mL) | A, B2 (80%), 8–10° | A, B2 (80%), 8–10° | C2 or C3 | A, B2$^{\ddagger}$ (80%), 8–10° | A, B2 (80%), 8–10° | C2 or C3 | |
| 26 | IDE | A2, B2 or B3 and B4 and B5 (50%), 0° | A2, B1, B2 or B3 and B4 and B5 (50%), 5° | C3 | A2, B2 or B3 and B4 and B5$^{\ddagger}$ (50%), 0° | A2, B1, B2 or B3 and B4 and B5 (50%), 2° | C3 | Note 1. |
| 27* | IDE | A, B2 or B3 (95%), 0–4° | – | C2 | A, B2 or B3 (95%), 0–4° | – | C2 | |
| 28 | Small, Helical Protein | A, B2 or B3 (80%), 5° | A, B2 or B3 (70%), 3° | C3 | A, B2 or B3$^{\ddagger}$ (80%), 5° | A, B2 or B3 (70%), 7° | C3 | |
| 29 | 300 kDa Protein | A or A2, B2 or B3 (70%), 7° | A or A2, B2 or B3 (50%), 13° | C3 | A or A2, B2 or B3$^{\ddagger}$ (70%), 7° | A or A2, B2 or B3 (50%), 9° | C3 | |
| 30*† | GDH | A, B3 (70%), 10° | A, B1, B3 (50%), 1° | C2 | A, B3$^{\ddagger}$ (70%), 10° | A, B1, B3 (50%), 16° | C3 | Note 2. Some non-adsorbed particles stack between layers. |
| 31*† | GDH | A, B3 (40%), – | A, B1, B3 (40%), 10° | C3 | A, B3$^{\ddagger}$ (40%), – | A, B1, B3 (40%), 2° | C2 | |
| 32*† | GDH (2.5 mg/mL)+0.001% DDM | A, B3 (40%), 4° | A, B1, B3 (40%), 7° | C2 | A, B3$^{\ddagger}$ (30%), 4° | A, B1, B3 (30%), 6° | C3 | Some non-adsorbed particles stack between layers. |
| 33*† | DnaB Helicase-helicase Loader | A, B2 or B3 (90%), 1° | A, B2 or B3 (90%), 4° | C3 | A, B2 or B3 (<5%), 1° | A, B2 or B3 (<5%), 1° | C2 | Gold flakes from Quantifoil are on the top. |
| 34*† | Apoferritin | A2, B2 or B3 (50%), 4–6° | – | C2 or C3 | A2, B2 or B3$^{\ddagger}$ (50%), 4–6° | – | C2 or C3 | Note 1. Note 2. |
| 35*† | Apoferritin | A2, B2 or B3 (60%), 4–12° | – | C2 or C3 | A2, B2 or B3$^{\ddagger}$ (60%), 4–12° | – | C2 or C3 | Note 1. Note 2. |
| 36*† | Apoferritin | A2, B3 (50%), 5° | A2, B1, B3 (50%), 10° | C3 | A2, B3$^{\ddagger}$ (70%), 5° | A2, B1, B3 (60%), 3° | C3 | Note 1. Note 2. |
| 37*† | Apoferritin (1.25 mg/mL) | A2, B2 or B3 (50%), 4–7° | A2, B1, B2 or B3 (50%), 6° | C3 | A2, B2 or B3$^{\ddagger}$ (40%), 4° | A2, B1, B2 or B3 (30%), 4° | C3 | Note 1. Note 2. |

Table 2 continued on next page

*Table 2 continued*

| Sample # | Sample name | Air-water interface, particle behavior, and layer/ice angle (bottom, center) | Air-water interface, particle behavior, and layer/ice angle (bottom, edge) | Ice behavior (bottom) | Air-water interface, particle behavior, and layer/ice angle (top, center) | Air-water interface, particle behavior, and layer/ice angle (top, edge) | Ice behavior (top) | Notes |
|---|---|---|---|---|---|---|---|---|
| 38*† | Apoferritin (0.5 mg/mL) | A2, B2 or B3 (20%), 5° | – | C2 or C3 | A2, B2 or B3‡ (20%), 1° | – | C2 or C3 | Note 1. Note 2. |
| 39*† | Apoferritin with 0.5 mM TCEP | A, B2 or B3 (40%), – or A, B2 or B3 (50%), 3° | A, B1, B2 or B3 (40%), 5–9° | C3 | A, B2 or B3 (40%), – or A, B2 or B3‡ (50%), 3° | A, B1, B2 or B3 (40%), 2–8° | C3 | Note 1. Note 2. |
| 40 | Protein with Carbon Over Holes | Carbon, B1 (30%), B3 (60%), 5° | Carbon, B1 (30%), B3 (60%), 5–9° | C2 | A, B3 (5%), 5° | A, B3 (5%), 5° | C1 or C2 | Note 3. |
| 41 | Protein and DNA Strands with Carbon Over Holes | A, No particles, 2–3° | – | C2 or C3 | Carbon, B1 (20%), B3 (60%), 2–3° | – | C2 | Some non-adsorbed particles make contact with particle layer. Most non-adsorbed particles are attached to DNA strands. |
| 42*† | T20S Proteasome | A, B3 (80%), 3° | A, B1 (5%), B3 (80%), 14° | C3 | A, B3‡ (80%), 3° | A, B1 (5%), B3 (20%), 3° | C2 | Note 2. Note 3. |
| 43*† | T20S Proteasome | A, B3 (10%), 2–5° | A, B3 (10%), 2–5° | C2 | A, B1 (20%), B3 (90%), 5–7° | A, B1 (20%), B3 (95%), 5–7° | C3 | Note 3. |
| 44*† | T20S Proteasome | A, B1 (10%), B3 (80%), 11° | – | C3 | A, B3 (2%), 11° | – | C2 | Note 2. Note 3. |
| 45*† | Mtb 20S Proteasome | – | A, B1, B2 or B3 (30%), 6° | C3 | – | A, B1, B2 or B3 (30%), 11° | C3 | Heavy contamination. |
| 46 | Protein on Streptavidin | Streptavidin, B2 (10–30%), 0° or Streptavidin, No particles, 12° | Streptavidin or A2, 2 (10–30%), 12° | C1, C2, or C3 | Streptavidin, B2 (10–30%), 0° or Streptavidin‡, No particles, 12° | Streptavidin, 2 (10–30%), 13–14° | C1, C2, or C3 | Note 1. Some holes have a layer of streptavidin only on top, some have a layer on top and bottom. Particles are attached to streptavidin and sometimes the apposed air-water interface. |

*A video is included for sample.

†A dataset is deposited for sample.

Note 1: Apparent protein fragments/domains are adsorbed to the air-water interfaces.

Note 2: Partial particles exist.

Note 3: Non-adsorbed particles make contact with particle layer.

DOI: https://doi.org/10.7554/eLife.34257.005

understand pathological particle behavior, and the use of fiducial-less cryoET for isotropic de novo model generation.

## Determination of tomography collection locations

The single particle samples studied here were sourced from a diverse set of grids, samples, and preparation techniques. Grid substrates include carbon and gold holey films, either lacey or with a variety of regularly spaced holes, and various nanowire grids (*Razinkov et al., 2016*) prepared using Spotiton. Grid types also include carbon Quantifoil (*Ermantraut et al., 1998*), gold Quantifoil (*Russo and Passmore, 2014*), and C-flat carbon on metal (*Quispe et al., 2007*). Plunging methods include plunging manually, with a Vitrobot (FEI Company, Hillsboro, OR) or CP3 (Gatan, Inc., Pleasanton, CA), and with Spotiton (*Jain et al., 2012*). With such diversity in samples and preparation techniques, we determined that the most feasible and representative collection strategy for analyzing particle and ice behaviors over dozens of preparations would be to collect in areas typical of where the sample owner intended to collect or had already collected single particle micrographs. For a typical grid, a low-magnification grid atlas or montage is collected, promising squares are

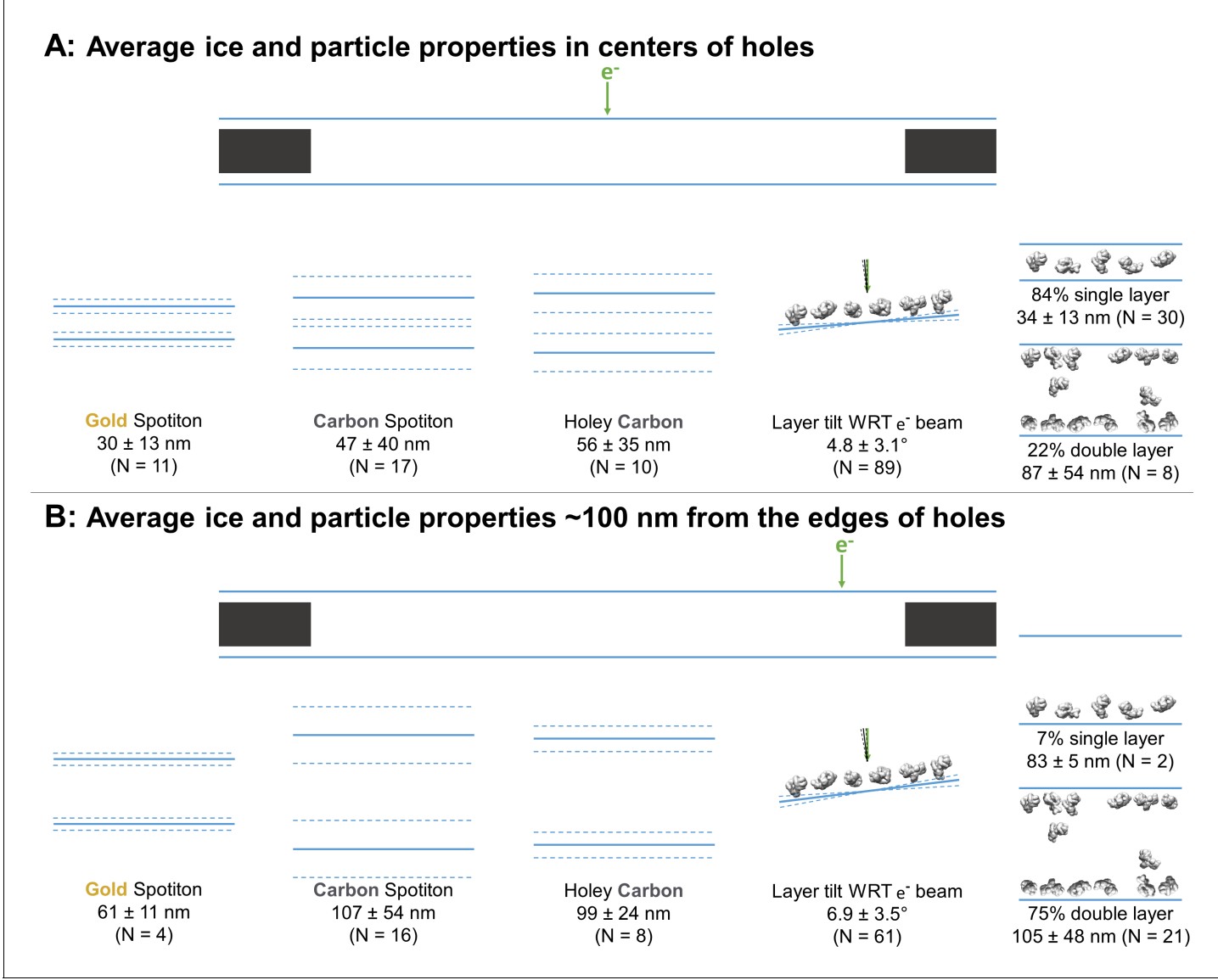

**Figure 3.** Schematic diagrams of the average ice thickness (solid lines) ± (1 standard deviation and measurement error) (dashed lines) using the minimum measured values, average particle layer tilt (solid lines) ± (1 standard deviation and measurement error) (dashed lines), and percentage of samples with single and/or double particle layers ('1' and/or '2' as defined in *Table 1*) at the centers of holes (**A**) and about 100 nm from the edge of holes (**B**).

DOI: https://doi.org/10.7554/eLife.34257.006

The following source data is available for figure 3:

**Source data 1.** Ice thickness and angle measurements for *Figure 3*.

DOI: https://doi.org/10.7554/eLife.34257.007

imaged at increasing magnifications, and potential exposure locations are examined at high magnification until sufficient particle contrast and concentration is found as determined by the sample owner. Then before or after a single particle collection, typically three or more tilt-series are collected as described in the Materials and methods. Tilt-series were typically collected from −45° to 45° with a tilt increment of 3°, defocus of ~5 microns, total dose of ~100 e-/Å$^2$, and a pixelsize between 1 and 2 Å. For most grids, one or two tilt-series are collected at the center of a typical hole and one or two tilt-series are collected at the edge of a typical hole, often including the edge of the hole if the grid substrate is carbon. Tilt-series are then aligned with Appion-Protomo (*Noble and Stagg, 2015*; *Winkler and Taylor, 2006*) for analysis as described in the Materials and methods.

| Sample # Name | Example cross-sectional schematic diagram | Sample # Name | Example cross-sectional schematic diagram | Sample # Name | Example cross-sectional schematic diagram | Sample # Name | Example cross-sectional schematic diagram |
|---|---|---|---|---|---|---|---|
| 1*<br>32 kDa Kinase | | 14*<br>Neural Receptor | | 27*<br>IDE | | 38*†<br>Apoferritin (0.5 mg/mL) | |
| 4*†<br>Hemagglutinin | | 17*<br>Protein with Bound Lipids (deglycosylated) | | 30*†<br>GDH | | 39*†<br>Apoferritin with 0.5 mM TCEP | |
| 5*<br>HIV-1 Trimer Complex 1 | | 18<br>Protein with Bound Lipids (glycosylated) | | 31*†<br>GDH | | 40<br>Protein with Carbon Over Holes | |
| 6*<br>HIV-1 Trimer Complex 1 | | 19*<br>Lipo-protein | | 32*†<br>GDH + 0.001% DDM (2.5 mg/mL) | | 41<br>Protein and DNA Strands with Carbon Over Holes | |
| 7*<br>HIV-1 Trimer Complex 2 | | 20<br>GPCR | | 33*†<br>DnaB Helicase-helicase Loader | | 42*†<br>T20S Proteasome | |
| 10*<br>Stick-like Protein 1 | | 21*†<br>Rabbit Muscle Aldolase (1mg/mL) | | 34*†<br>Apoferritin | | 43*†<br>T20S Proteasome | |
| 12*<br>Stick-like Protein 2 | | 22*†<br>Rabbit Muscle Aldolase (6mg/mL) | | 35*†<br>Apoferritin | | 44*†<br>T20S Proteasome | |
| 13*<br>Neural Receptor | | 25*<br>Protein in Nanodisc (0.58 mg/mL) | | 36*†<br>Apoferritin | | 45*†<br>Mtb Proteasome | |
| | | | | 37*†<br>Apoferritin (1.25 mg/mL) | | 46<br>Protein on Streptavidin | |

**Figure 4.** A selection of cross-sectional schematic diagrams of particle and ice behaviors in holes as depicted according to analysis of individual tomograms. The relative thicknesses of the ice in the cross-sections are depicted accurately. Each diagram is tilted corresponding to the tomogram from which it is derived; i.e. the depicted tilts represent the orientation of the objects in the field of view at zero-degree nominal stage tilt. If the sample concentration in solution is known, then it has been included below the sample name. Black lines on schematic edges are the grid film. The cross-sectional characteristics depicted here are not necessarily representative of the aggregate. An asterisk (*) indicates that a Video of the schematic diagram alongside the corresponding tomogram slice-through video is included for the sample. A dagger (†) indicates that a dataset is deposited for sample. A generic particle, holoenzyme EMDB-6803 (*Yin et al., 2017*), is used in place of some confidential samples (samples #40, 41, and 46).
DOI: https://doi.org/10.7554/eLife.34257.008

## Analysis of single particle tomograms

Single particle tomograms of samples described in *Table 1* have each been analyzed visually using 3dmod from the IMOD package (*Kremer et al., 1996*). After orienting a tomogram such that one of the air-water interfaces is approximately parallel to the visual plane, traversing through the slices of the tomogram allows for the determination of relative particle locations, orientations, ice thickness variations in holes, and measurement of the minimum particle distance from the air-water interfaces. For many of the samples shown here and made available in the data depositions, particle orientations can be explicitly determined by direct visualization. Contamination on the surface of the air-water interface is used to determine the approximate location of the interface and to measure the ice thicknesses. After analyzing hundreds of single particle tomograms, we have concluded that sequestered layers of proteins in holes always correspond to an air-water interface,

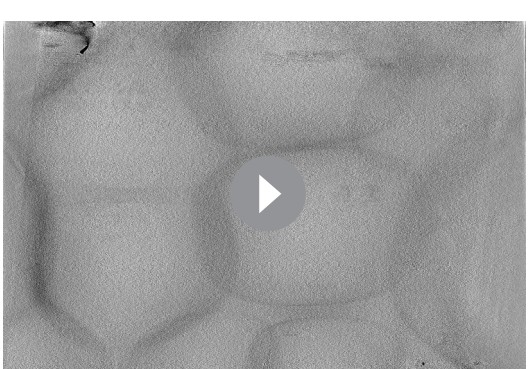

**Video 1.** Sample 20.
DOI: https://doi.org/10.7554/eLife.34257.010

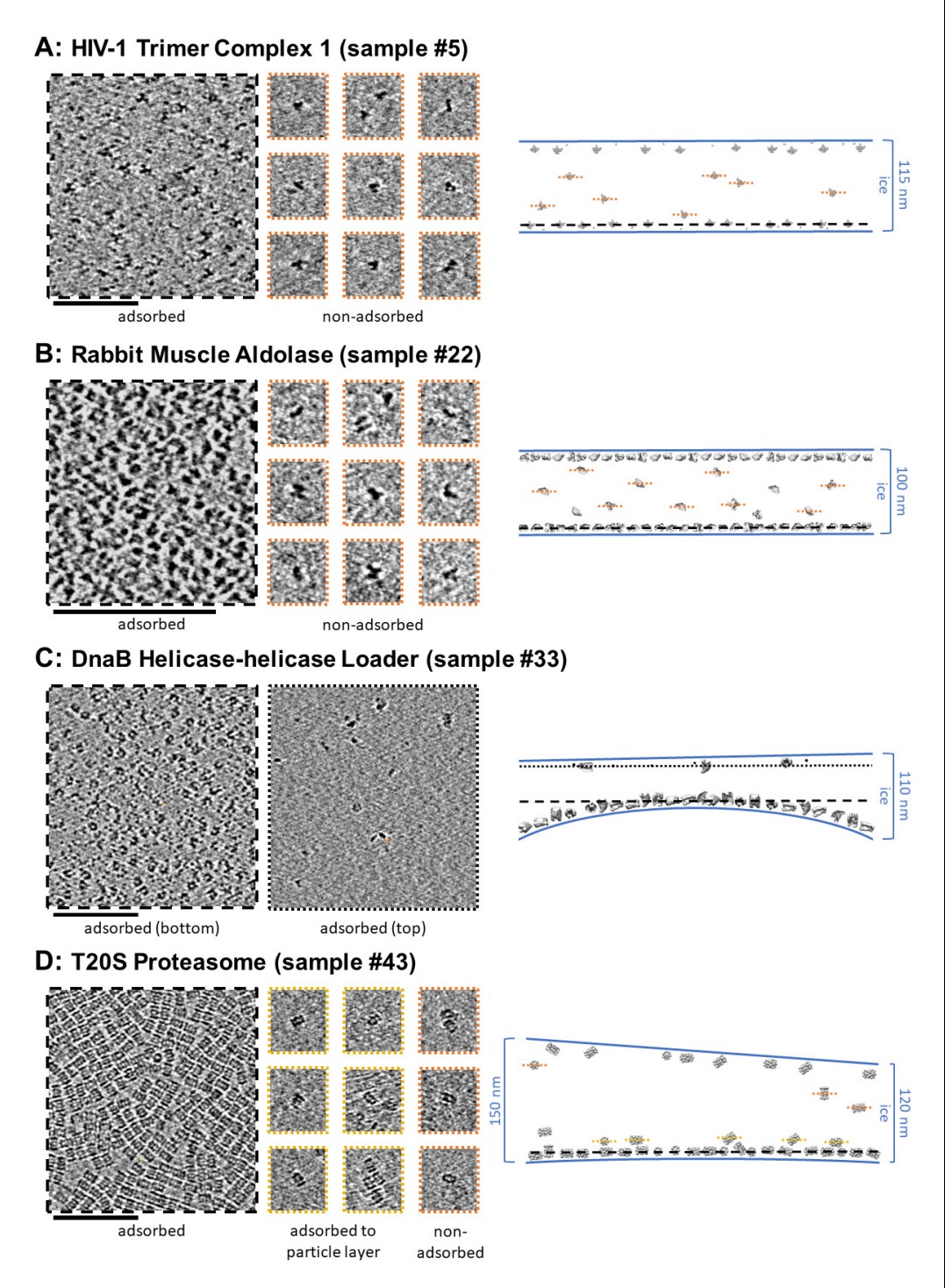

**Figure 5.** Slices of tomograms, about 7 nm thick, showing variations in particle orientation of adsorbed and non-adsorbed particles for several samples. Cross-sectional schematic diagrams showing the approximate locations of the slices are shown on the right. (**A**) HIV-1 trimer complex 1 shows a high degree of preferred orientation for particles adsorbed to the air-water interface and no apparent preferred orientation for non-adsorbed particles. (**B**) Rabbit muscle aldolase shows several views for adsorbed particles and non-preferred views for non-adsorbed particles. (**C**) DnaB helicase-helicase loader shows no apparent preferred orientation for adsorbed particles. (**D**) T20S proteasome shows predominantly one view for adsorbed particles, the same view for particles adsorbed to the primary layer of particles, and less preferred views for non-adsorbed particles. Scale bars are 100 nm.
DOI: https://doi.org/10.7554/eLife.34257.009

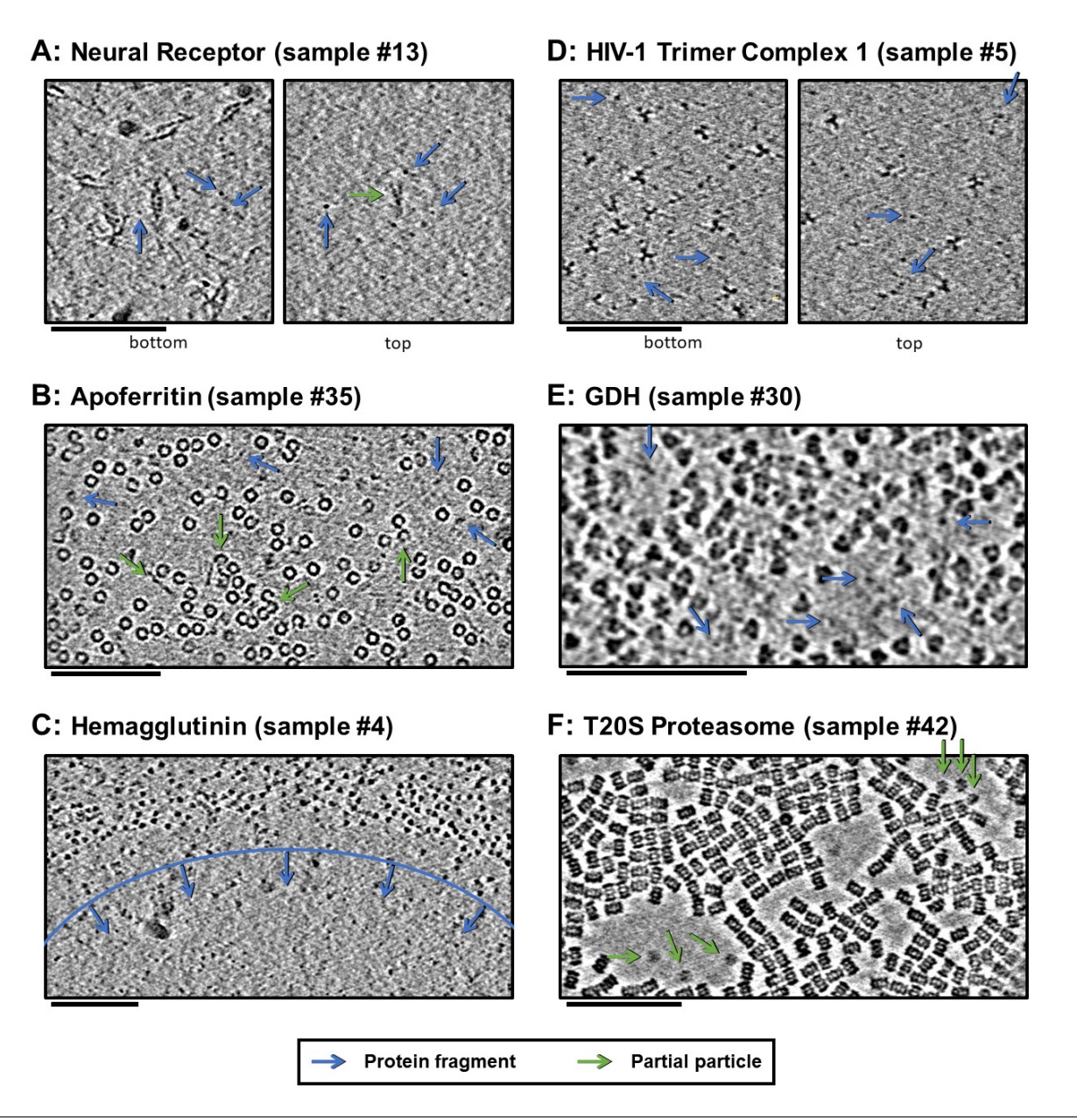

**Figure 6.** Slices of tomograms, about 10 nm thick, at air-water interfaces of samples that show clear protein fragments (examples indicated with blue arrows) and/or partial particles (examples indicated with green arrows), presented roughly in order of decreasing overall fragmentation. (A) Neural receptor shows a combination of fragmented 13 kDa domains consisting primarily of β-sheets and partial particles. (B) Apoferritin shows apparent fragmented strands and domains along with partial particles. (C) Hemagglutinin shows a clear dividing line, marked with blue, where the ice became too thin to support full particles, but thick enough to support protein fragments. (D) HIV-1 trimer complex one shows several protein fragments on the order of 10 kDa; however, these might be receptors intentionally introduced to solution before plunge-freezing. (E) GDH shows protein fragments interspersed between particles. (F) T20S proteasome shows partial particles, determined by measuring their heights in the z-direction, on an otherwise clean air-water interface (see the end of *Video 10* for sample #42). For the examples shown here, it is not clear whether the protein fragments and partial particles observed are due to unclean preparation conditions, protein degradation in solution, or unfolding at the air-water interfaces, or a combination; all cases are expected to result in the same observables due to competitive and sequential adsorption. Scale bars are 100 nm.
DOI: https://doi.org/10.7554/eLife.34257.011

thus providing a second method for determining the location of the interface.

*Table 1* is organized with the single particle sample mass in roughly descending order. Over 1000 single particle tomograms of over 50 different sample preparations have been collected over a 1-

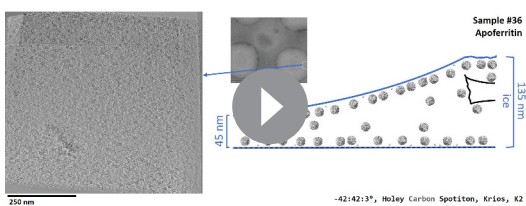

**Video 2.** Sample 34.
DOI: https://doi.org/10.7554/eLife.34257.012

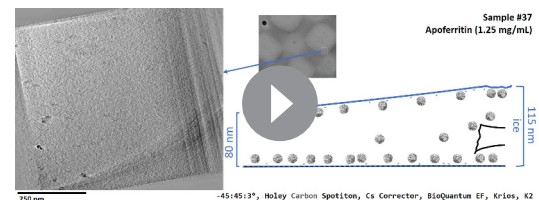

**Video 3.** Sample 35.
DOI: https://doi.org/10.7554/eLife.34257.013

year period. Most of these samples are reported on here. These samples include widely studied specimen such as glutamate dehydrogenase (GDH), apoferritin, and T20S proteasome (samples #30–32, #34–39, and #42–44, respectively), along with various unique specimens such as a neural receptors, lipo-protein, and particles on affinity grids (samples #13,14, #19, and #40, 41, 46, respectively). Samples that are not specifically named have yet to be published. Over half of the samples were prepared on gold or carbon nanowire grids, while the remaining were prepared on a variety of carbon and gold holey grids using common cryo-plunging machines and techniques. Samples showing regions of ice in grid holes with near-ideal conditions – less than 100 nm ice thickness, no overlapping particles, and little or no preferred orientation – are highlighted in blue (21 of 46 samples; 46%) in *Tables 1* and *2*. Samples showing regions of ice in grid holes with ideal conditions – near-ideal conditions plus no particle-air-water interface interaction – are highlighted in green (2 of 46 samples; 4%). Over half of the samples only contained areas that are not ideal for collection due to ice thickness being greater than 100 nm, overlapping particles, and/or preferred orientation.

## Ice thickness

Averages ± (one standard deviation and measurement error) of the minimum ice thickness at the center and near the edge of grid holes was calculated. At the center, the ice thickness is about 30 ± 13 nm for gold nanowire grids prepared with Spotiton (N = 11), 47 ± 40 nm for carbon nanowire grids prepared with Spotiton (N = 17), and 56 ± 35 nm for carbon holey grids prepared using conventional methods (N = 10) (*Figure 3A*). Ice thickness about 100 nm from the edge of grid holes is about 61 ± 11 nm for gold nanowire grids prepared with Spotiton (N = 4), 107 ± 54 nm for carbon nanowire grids prepared with Spotiton (N = 16), and 99 ± 24 nm for carbon holey grids prepared using conventional methods (N = 8) (*Figure 3B*).

*Table 2* categorizes each sample in terms of *Figure 2*. Categorizations into A, B, and C, where possible, have been judged by visual inspection. Air-water interfaces that are visually clean are denoted with 'A' from *Figure 2* due to A1, A2 (primary structure), and A3 being indistinguishable by cryoET without collecting high tilt angles, which was not done in this study. For particles smaller than about 100 kDa, distinguishing between A1/A3 and A2 was not possible by cryoET.

If a region in grid holes contains layers of particles relative to the air-water interface (possibly B1 – B4), then the particle saturation of the corresponding layer is recorded in *Table 2* as an approximate percentage in parentheses where 100% means that no additional particles could be fit into the layer. The angle of particle layer with respect to the electron beam is recorded for each region if applicable. The average tilt ± (one standard deviation and measurement error) of layers at the

**Video 4.** Sample 36.
DOI: https://doi.org/10.7554/eLife.34257.014

**Video 5.** Sample 37.
DOI: https://doi.org/10.7554/eLife.34257.015

**Video 6.** Sample 38.
DOI: https://doi.org/10.7554/eLife.34257.016

**Video 7.** Sample 04.
DOI: https://doi.org/10.7554/eLife.34257.017

centers of holes is 4.7 ± 3.0° and at the edges of holes is 6.9 ± 3.5° (*Figure 3*). There is no apparent correlation between microscope and tilt direction or magnitude. About 83% of the samples contained single particle layers (N = 30) in the centers of holes while about 22% contained double particle layers (N = 8; several samples have different holes with single and double layers of particles in their centers). Near the edges of holes, about 7% contain single particle layers (N = 2), while about 75% contained double particle layers (N = 21). Finally, in *Table 2* the ice curvature of each air-water interface is specified using the options in *Figure 2C*. For these measurements, the bottom of each tomogram is defined as having a lower z-slice value than the top as viewed in 3dmod, yet the relative orientation of each recorded sample is not known due to unknown sample application orientation on the grid relative to the EM stage. Thus, correlations between air-water interface behavior and sample application direction on the grids cannot be made from this study.

## Cross-sectional depictions

Several schematic diagrams of cross-sections of particle and ice behavior in holes as determined by cryoET are shown in *Figure 4* for selected samples and tomograms. Ice thickness measurements and particle sizes are approximately to scale. Each cross-section is tilted corresponding to the tilt of the tomogram from which it was derived relative to the electron beam. The preferred orientation distributions are reflected in the cross-sectional depictions. The cross-sectional characteristics depicted are not necessarily representative of the average because only one of several collected tomograms are depicted.

Several tomographic slice-through videos from representative imaging areas of samples are shown in the included Videos. Most of the Videos include the corresponding hole magnification image, which is an order of magnitude lower magnification than exposure magnification, with the location of the targeted area specified. Tilt-series collection range, grid type, and collection equipment are also specified. Tomography may also be performed at hole magnification, allowing for particle location determination across multiple sized holes, ice thickness determination, and local grid tilt (*Video 1* - sample #20). For sample #20, a GPCR with a particle extent of about 5 nm, a

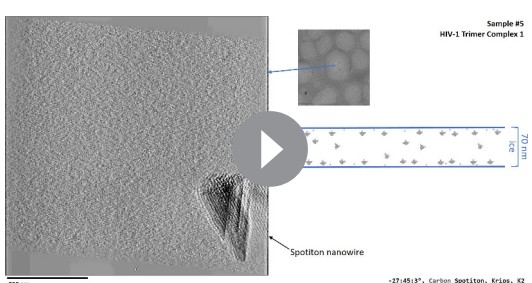

**Video 8.** Sample 05.
DOI: https://doi.org/10.7554/eLife.34257.018

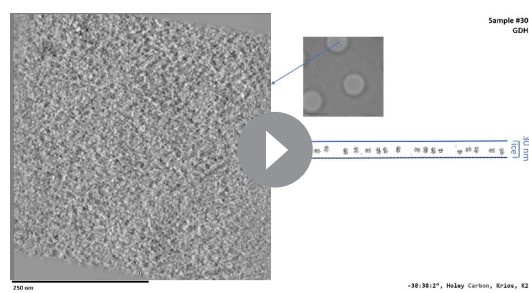

**Video 9.** Sample 30.
DOI: https://doi.org/10.7554/eLife.34257.019

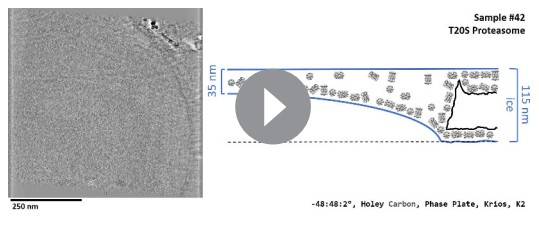

**Video 10.** Sample 42.
DOI: https://doi.org/10.7554/eLife.34257.020

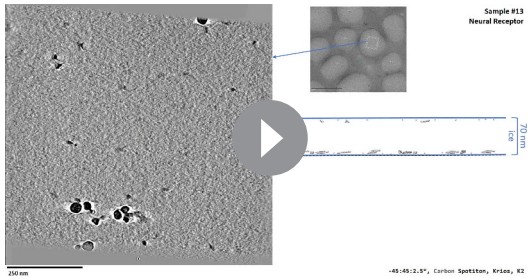

**Video 11.** Sample 13.
DOI: https://doi.org/10.7554/eLife.34257.021

**A: Limits imposed by ice thickness variations**

Estimated defocus of
whole projection image

Harsh
limit

Minimum
limit

Resolution limit based on
whole projection image
defocus estimation

**B: Additional effects imposed by particle layer tilts**

**Figure 7.** Collection and processing limits imposed by variations in ice thickness (**A**) and particle layer tilt (**B**), given that the vast majority of particles in holes on conventionally-prepared cryoEM grids are adsorbed to an air-water interface. (**A**) Variations in ice thickness within and between holes might limit the number of non-overlapping particles in projection images (efficiency of collection and processing), the accuracy of whole image and local defocus estimation (accuracy in processing), the signal-to-noise ratio in areas of thicker ice (efficiency of collection and processing), and the reliability of particle alignment due to overlapping particles being treated as a single particle. (**B**) Variations in the tilt angle of a given particle layer might affect the accuracy of defocus estimation if the field of view is not considered to be tilted, yet will increase the observed orientations of the particle in the dataset if the particle exhibits preferred orientations. Dashed black lines indicate the height of defocus estimation on the projected cross-section if sample tilt is not taken into account during defocus estimation. Particles are colored relative to their distance from the whole image defocus estimation to indicate the effects of ice thickness and particle layer tilt. Gray particles would be minimally impacted by whole-image CTF correction while red particles would be harshly impacted by whole-image CTF correction. Particles that would be uniquely identifiable in the corresponding projection image are circled in green.
DOI: https://doi.org/10.7554/eLife.34257.023

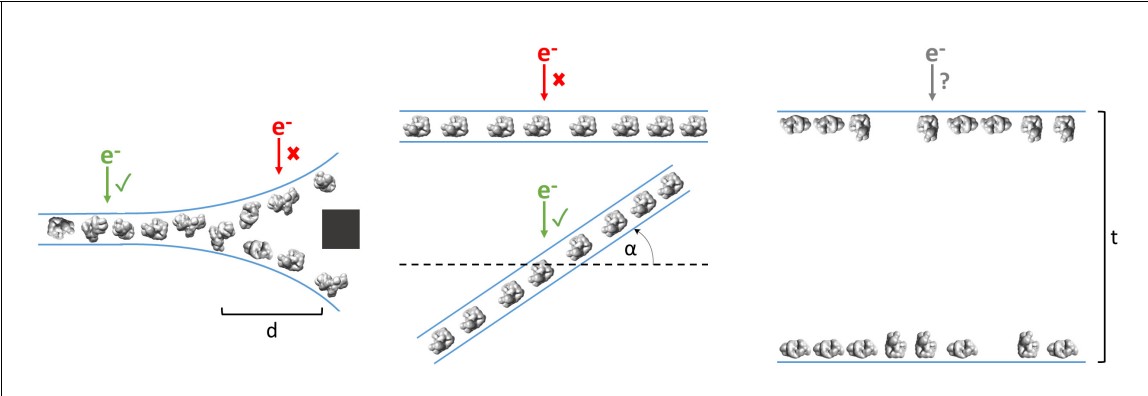

**Figure 8.** Examples of typical single particle and ice behavior as might be revealed by fiducial-less cryoET and how such characterization might influence strategies for single particle collection. Left: For a sample that exhibits thick ice near the edges of holes and ice in the center of holes that is thin enough for a single layer of particles to reside, single particle micrographs would optimally be collected a distance, d, away from the edges of holes. Middle: A sample that exhibits a high degree of preferred orientation may require tilted single particle collection by intentionally tilting the stage by a set of angles, α, in order to recover a more isotropic set of particle projections (*Tan et al., 2017*). Right: For a sample that consists of multiple layers of particles across holes, the sample owner may decide to proceed with collection with the knowledge that the efficiency will be limited by the particle saturation in each layer and that the resolution will be limited by the decrease in signal due to the ice thickness, t, and the accuracy of CTF estimation and correction. The results of cryoET on a given single particle cryoEM grid might also result in the sample owner deciding that the entire grid is not worth collecting on, potentially due to the situations described here or due to observed particle degradation. Due to depiction limitations, the single orientation of the particle in the middle column is depicted as being only in one direction, when in practice the particles may rotate on the planes of the air-water interfaces.

DOI: https://doi.org/10.7554/eLife.34257.024

tomographic analysis at hole magnification (about 20 Å pixelsize) is sufficient to localize ice contamination, particle layers, and to measure ice thickness with an accuracy of about 10 nm. To orient the reader to this single particle tomography data, *Figure 5* shows tomogram slice-throughs of adsorbed and non-adsorbed particles for a selection of samples with thicker ice.

## The vast majority of particles are localized to the air-water interfaces

The primary result gleaned from over 1000 single particle tomograms of over 50 different grid/sample preparations is that the vast majority of all particles (approximately 90%) are local to an air-water interface. As shown in *Table 1*, *Table 2*, *Figure 4*, and the Videos, most particles prepared with sample incubation times on the order of 1 s on the grid are within 5–10 nm of an air-water interface (ie. are characterized by B2, B3, or B4 in *Table 2*). This observation implies that most particles, not only in this study but in cryoEM single particle studies as a whole, are adsorbed to an air-water interface.

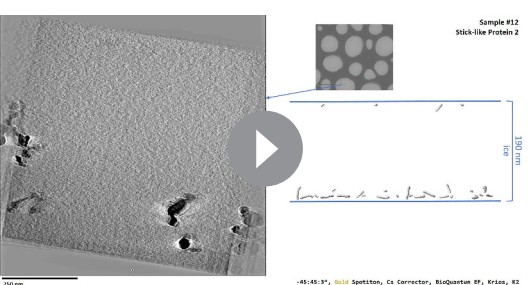

**Video 12.** Sample 12.
DOI: https://doi.org/10.7554/eLife.34257.022

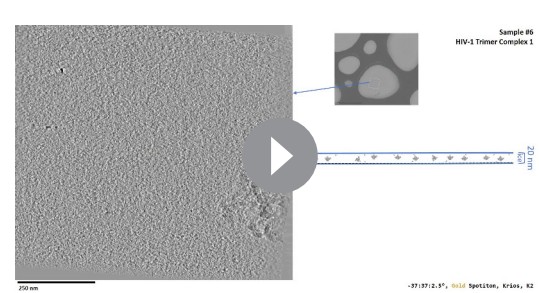

**Video 13.** Sample 6.
DOI: https://doi.org/10.7554/eLife.34257.025

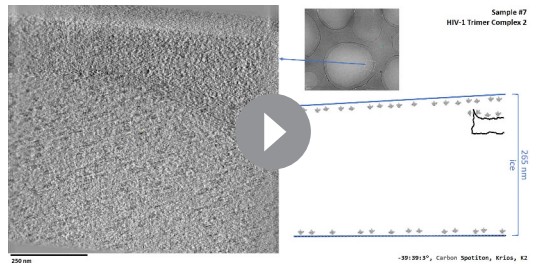

**Video 14.** Sample 7.
DOI: https://doi.org/10.7554/eLife.34257.026

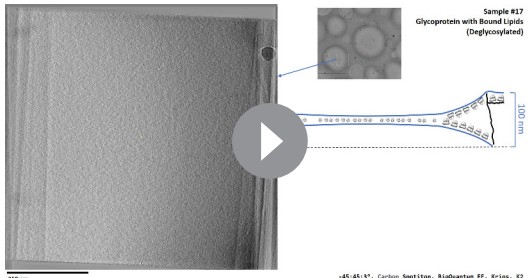

**Video 15.** Sample 17.
DOI: https://doi.org/10.7554/eLife.34257.027

## A: Gaussian particle picking

2D classification

Incoherent *ab initio*
reconstruction

## B: CryoET SPT produces *de novo* templates for picking and alignment

Five tomograms;
~1,000 particles

SPT

45°

45°

template picking

2D classification

Initial model

4.1 Å isotropic
reconstruction

**Figure 9.** De novo initial model from f ducial-less SPT. (**A**) Gaussian picking of single particle datasets of DnaB helicase-helicase loader was not able to identify many low contrast side-views of the particle and 2D classification of the top-views incorrectly suggested C6 symmetry, resulting in unreliable initial model generation and stymying efforts to process the datasets further. (**B**) Fiducial-less single particle tomography (SPT) on the same grids used for single particle collection was employed to generate a de novo initial model, which was then used both as a template for picking all views of the particle in the single particle micrographs and as an initial model for single particle alignment, resulting in a 4.1 Å isotropic structure of DnaB helicase-helicase loader (manuscript in preparation). This exemplifies the novelty of applying this potentially crucial fiducial-less SPT workflow on cryoEM grids. Scale bars are 100 nm for the micrographs and tomogram, 10 nm for the 2D classes, and 5 nm for the 3D reconstructions.
DOI: https://doi.org/10.7554/eLife.34257.028

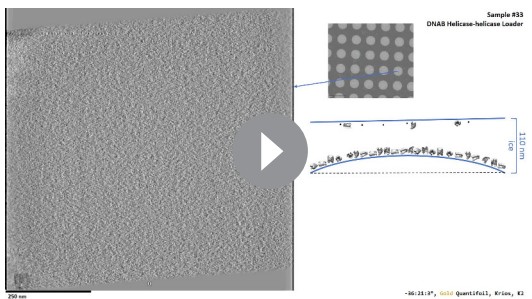

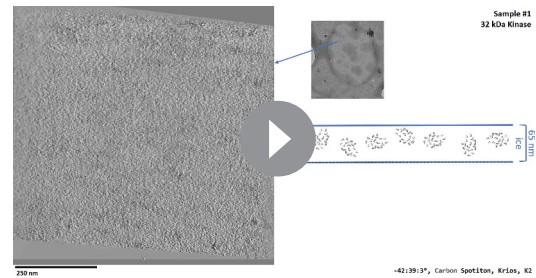

**Video 16.** Sample 33.
DOI: https://doi.org/10.7554/eLife.34257.029

**Video 17.** Sample 01.
DOI: https://doi.org/10.7554/eLife.34257.030

## Particle adsorption sometimes implies preferred orientation

A sequestered particle that is adsorbed to a clean air-water interface and that has had time to equilibrate will likely be oriented relative to that air-water interface such that the local surface hydrophobicity of the particle is maximally exposed, assuming that the particle is not prone to denaturation at the interface. If a particle is prone to denaturation at the interface and if the interface is already coated with a denatured layer of protein, then the preferred orientations of the same sequestered particle on the protein film-air-water interface might change. If the particle is not sequestered, but is in a protein-concentrated environment, then neighboring particle-particle interactions might change the possible preferred orientations of the particles. For each of these cases, an ensemble of particles at air-water interfaces arrived at by diffusion, as is the case with most single particle cryoEM datasets, will exhibit all possible particle orientations. The percentage of particles in each preferred orientation might be then mapped back onto all possible relative local particle-air-water interface affinities. Particles that have had less time to equilibrate before observation (e.g. before plunge-freezing) might have more realized orientations in the ensemble than if they had more time to equilibrate at the air-water interfaces. Several example tomogram slice-throughs of samples with varying amounts of apparent preferred orientations at and away from air-water interfaces are shown in *Figures 5* and *6*.

Protein adsorption to an air-water interface has potential consequences with regard to protein denaturation, data collection, and image processing. In the remainder of this section, we will discuss the implications of protein adsorption on protein denaturation and present possible evidence of air-water interface denaturation from cryoET.

## Observed denatured proteins by cryoET

Several samples show clear protein fragments at air-water interfaces (samples #4–6, 10–14, 26, 30, 34–38, and 46; *Figure 6A–E*, blue arrows). The neural receptor, hemagglutinin, HIV-1 trimer complex 1, apoferritin, and GDH samples in particular (samples #13, #35, #4, #5, and #30, respectively) show protein fragments and domains on the air-water interfaces (*Figure 6A–E* and corresponding Videos, blue arrows). For the neural receptors (sample #13), densities on the air-water interface show a clear

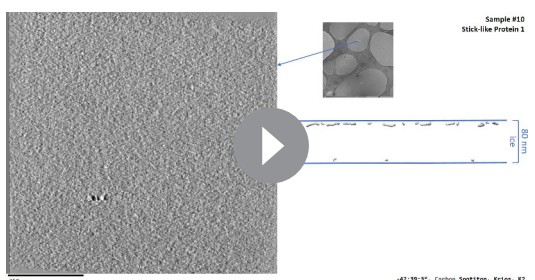

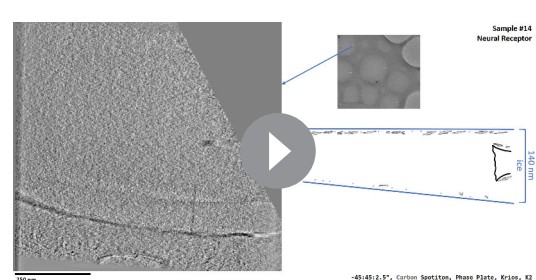

**Video 18.** Sample 10.
DOI: https://doi.org/10.7554/eLife.34257.031

**Video 19.** Sample 14.
DOI: https://doi.org/10.7554/eLife.34257.032

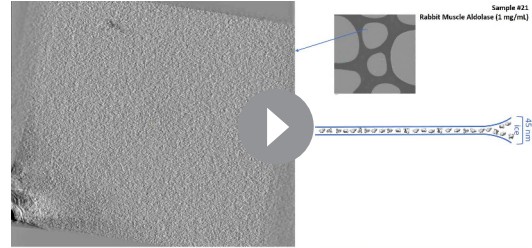

**Video 20.** Sample 19.
DOI: https://doi.org/10.7554/eLife.34257.033

**Video 21.** Sample 21.
DOI: https://doi.org/10.7554/eLife.34257.034

relationship in size to the 13 kDa Ig-like domains that constitute the proteins. Several apoferritin samples (samples #34–38) also show apparent protein fragments at the air-water interfaces (*Figure 6B* and *Videos 2–7* corresponding to samples #34–38). One hemagglutinin sample contained holes where the ice became too thin for whole particles to reside and is instead occupied exclusively by protein fragments (*Figure 6C* and *Video 7* corresponding to sample #4). An HIV-1 trimer sample also shows clear protein fragments on each air-water interface, although these are likely receptors intentionally introduced to solution before plunge-freezing (*Figure 6D* and *Video 8* corresponding to sample #5). GDH similarly shows sequestered protein fragments in open areas near particles at the air-water interface (*Figure 6E* and *Video 9* corresponding to sample #30).

Several samples show clear partial particles at air-water interfaces (samples 10–14, 34–39, and 46; *Figure 6A,B,F*, green arrows). Neural receptor (sample #13) particle fragments can be seen adsorbed to the air-water interface (*Figure 6A*, green arrow). Sample #13 consists of two distinct air-water interfaces, as can be seen in *Figure 6A*, where the bottom interface is covered with particles and protein fragments while the top interface is covered with protein fragments and a small number of partial particles (see also *Video 11* corresponding to sample #13). The partial T20S proteasome particles shown in *Figure 6F* and the *Video 10* (sample #42) might be an example of protein denaturation at the air-water interface. In this sample, the observed partial particles are oriented as rare top-views rather than abundant side-views of the particle and exist adjacent to areas of the air-water interface that do not harbor adsorbed particles. Also of note is that all of the domains of the neural receptor and some of the domains of apoferritin, hemagglutinin, HIV-1 trimer complex 1, and GDH are composed of series of β-sheets, which have the potential to not denature at the air-water interface. This observation might correlate with the cross-disciplinary literature presented in the introduction showing that β-sheets may potentially survive air-water interface interaction (*Martin et al., 2005*; *Renault et al., 2002*; *Yano et al., 2009*). It is unclear, however, whether these unclean air-water interfaces are due to unclean preparation conditions (*Glaeser et al., 2016*), protein degradation in solution, unfolding at the air-water interfaces, or a combination of these factors.

While the observations described above might correlate with the research from the food science and surface physics literature as outlined in the introduction, it is not clear from this study whether particles are adsorbed to films of denatured protein at the air-water interface or if some particles are adsorbed directly to the air-water interface. From the cross-disciplinary literature presented in the introduction, we speculate that adsorption rates for proteins that first denature at the air-water interface will differ from those that adsorb directly to the air-water interface. For a protein that does denature at the air-water interface, there is an additional amount of diffusion time, possibly on the order of tens of milliseconds, for surface diffusion to take place. Proteins that adsorb directly to the air-water interface are only time-limited by the bulk diffusion time of that

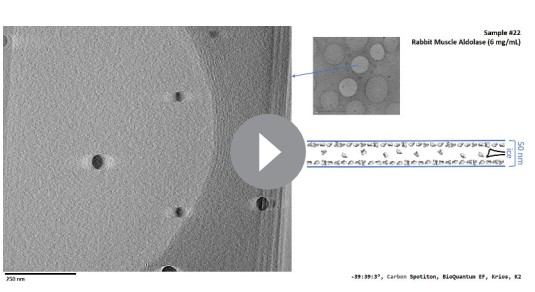

**Video 22.** Sample 22.
DOI: https://doi.org/10.7554/eLife.34257.035

sample preparation. The bulk diffusion time may be orders of magnitudes less than the surface diffusion time. The rate at which proteins adsorb to a protein network film depends on the affinity between that protein film and the bulk particles. The additional surface diffusion time along with the additional bulk protein adsorption time to the denatured protein film may allow for speed advances in sample application and plunging to outrun bulk protein adsorption to the denatured proteins on the air-water interfaces, depending on the grid preparation and particle behavior. Secondary effects, such as bulk particle flow – in conventional grid preparation when blotting paper is applied and in nanowire grid preparation with Spotiton when the protein solution reaches the nanowires on the grid bars and wicks away – and flow due to thermal convection – potentially due to contact with tweezers and the blotting process – may change the effective concentration of bulk particles near the air-water interfaces.

## Protein network films may not be particle-friendly

Evidence from the literature in the introduction shows that proteins do denature at air-water interfaces, with an apparent dependency on protein concentration and structural rigidity. Evidence from this study showing that some air-water interfaces do harbor protein fragments and/or partial particles might be additional examples of denaturation due to the air-water interface. Evidence from LB trough studies of the small, disordered protein β-casein additionally show that increasing the concentration of bulk proteins in solution from 0.1 to 100 mg/mL results in an increased thickness of the denatured protein film at the air-water interface from 5 to 50 nm (*Meinders et al., 2001*). This observation implies that bulk proteins may denature not only at the air-water interface, but also at the subsequently formed protein network film interface depending on the bulk protein concentration. This in turn implies that proteins adsorbed to the protein film undergo conformational change, at least at higher concentrations. Thus, if an increase in the thickness of a protein network film of a given protein at high concentration is observed, concern that bulk proteins adsorbed to the protein network film are undergoing conformational change might be warranted. We speculate that if particles are undergoing conformational change at either the protein-air-water interface or at the protein-protein network interface, then anomalous structures might be present after 2D and 3D classification that are practically indistinguishable from the nominal structures. These anomalous structures might contribute toward artefactual 3D reconstructions, towards lower resolutions, and/or toward lower density contributions on the peripheries of resulting 3D reconstructions. In the last two cases, lower resolutions on the peripheries of the reconstruction might also be a result of radial inaccuracies in alignment, and thus these two resolution-degrading factors would need to be decoupled on a per-sample basis before drawing conclusions. Apoferritin, as shown in *Figure 6B* and the *Videos 2–6* (samples #34–38), might be an explicit example of observed conformational change due to the air-water interface if the observed particle degradation is indeed caused by air-water interface denaturation.

## Air-water interface symmetries and asymmetries

Several samples show an asymmetry between particle saturation at the top and bottom air-water interfaces. For example, samples #10, 12–15, 33, and 44 have particles covering one air-water interface with the other interface showing no particles, samples #4, 7, 9, 18, 32, 36, 39, 42, and 43 have more particles covering one air-water interface than the other, and samples #1, 8, 9, 16, 17, 20–22, 24–31, 39, and 45 have a roughly equal number of particles on each air-water interface (*Figure 6*). Particles that layer only on one air-water interface suggest that they are either sticking to the first available air-water interface (the interface on the back of the grid prior to blotting for conventional grid preparation techniques or the interface in the direction of application momentum for Spotiton), or to the first-formed protein network film. This first-formed protein network film might form nearly instantaneously after the

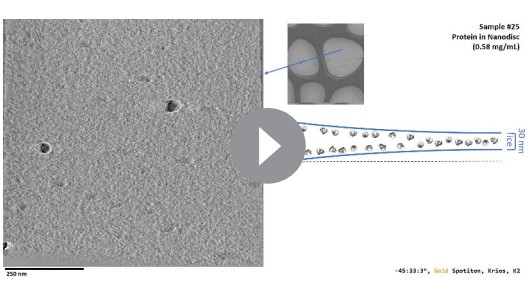

**Video 23.** Sample 25.
DOI: https://doi.org/10.7554/eLife.34257.036

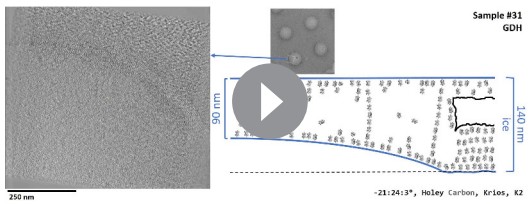

**Video 24.** Sample 27.
DOI: https://doi.org/10.7554/eLife.34257.037

**Video 25.** Sample 31.
DOI: https://doi.org/10.7554/eLife.34257.038

first air-water interface is created with the sample dispenser. For a particle that denatures at the air-water interface, since the bulk diffusion time is one or more orders of magnitude less than the surface diffusion time, if the second available air-water interface is formed before the first air-water interface is saturated with bulk particles and if the protein concentration is high enough, then one might expect denaturation to occur at the second air-water interface. This would allow for a layer of particles to adsorb to each air-water interface. Further study into such sample behavior using cryoET while taking into account sample application directionality might lead to a clearer model for why particles adsorb preferentially to one air-water interface over the other.

Ideal samples are a rarity

Only two samples, #25: protein in nanodisc and #46: protein on streptavidin, exhibit ideal characteristics – less than 100 nm ice thickness, no overlapping particles, little or no preferred orientation, and areas with no particle-air-water interface interaction. Sample #25 contains regions of single layers of particles in nanodiscs without preferred orientation in 30 nm ice (see corresponding *Video 12*). While the particle layers are on the air-water interfaces in thicker areas near the edges of holes, the lack of preferred orientation implies that some fraction of the particles contain protein that is not in contact with the air-water interface, thus satisfying the ideal condition. Sample #46 contains particles dispersed on streptavidin, which is used to both randomly orient the particles and to avoid at least one air-water interface (*Figure 4*). The majority of areas with particles consists of ice thin enough to satisfy the ideal condition.

A single particle dataset consisting primarily of adsorbed particles to air-water interfaces not only opens up the possibility of protein and degradation conformational change as described in this section, but additionally has implications on data collection and image processing as described in the next three sections.

## A significant fraction of areas in holes have overlapping particles in the electron beam direction

A large fraction of the samples studied here contain imaging areas in holes, often limited to near the edges of holes, contain a single layer of particles at an air-water interface with additional non-adsorbed particles or two layers of particles with or without additional non-adsorbed particles (denoted in *Table 1* as having 1+, 2, or 2+ layers in holes) (*Figure 3*). When this occurs, it is often the case that projection images collected in these areas will contain overlapping particles

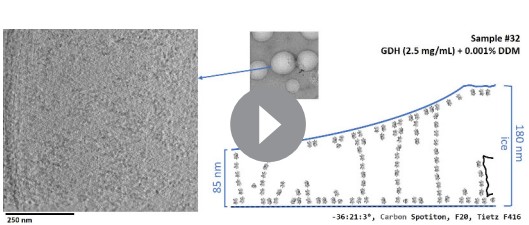

**Video 26.** Sample 32.
DOI: https://doi.org/10.7554/eLife.34257.039

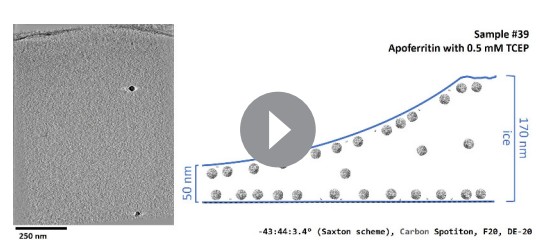

**Video 27.** Sample 39.
DOI: https://doi.org/10.7554/eLife.34257.040

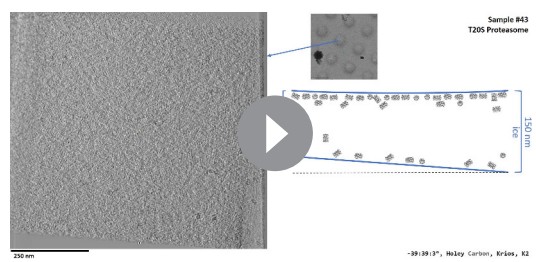

**Video 28.** Sample 43.
DOI: https://doi.org/10.7554/eLife.34257.041

(*Figure 7A*, middle and right). These overlapping particles may cause several issues. First, overlapping particles picked as one particle will need to be discarded during post-processing (particles not circled in *Figure 7*). If these particles are not discarded, then anomalous results might be expected in any 3D refinement containing these particles – particularly in refinement models that use maximum likelihood methods such as Relion (*Scheres, 2012*), cryoSPARC (*Punjani et al., 2017*), and Xmipp (*Scheres et al., 2007*; *Scheres et al., 2008*) – thus reducing the reliability and accuracy of the refinement results. Second, overlapping particles reduce the accuracy of whole-image defocus estimation (as depicted by particle color in *Figure 7*). For instance, an exposure area perpendicular to the electron beam containing two parallel layers of particles with identical concentrations will result in a whole-image defocus estimation located halfway-between the two layers, thus limiting the resolution of each particle depending on their distance from the midway point. For such an image collected with a defocus range of 1 to 2 microns and with a 10 nm deviation from the midway point, the particles will have a resolution limit of about 2.5 Å. A 50 nm deviation from the midway point will result in a resolution limit of about 6 Å. Third, overlapping particles might reduce the accuracy of per-particle or local defocus estimation. If the concentrations of overlapping particles are too high, then local and potentially per-particle defocus estimation might contain fragments of particles at different heights than the particle of interest. Fourth, overlapping particles reduce the efficiency of data processing and thus data collection. The second and the third issues posed above might be partially resolved if the ice thickness is known by duplicating each particle, CTF correcting one

with (midway defocus + thickness/2) and the other with (midway defocus – thickness/2), then discarding the particle with the lower high-frequency cross correlation value partway through single particle alignment. The issues posed above may be a primary source of discarded particles during mean filtering, CTF confidence filtering, 2D classification, and 3D classification.

## Most air-water interfaces are tilted with respect to the electron beam

We have shown that the majority of samples studied contain particles at one or both air-water interfaces (*Tables 1* and *2*, *Figure 3*). Tomography also has allowed us to study the orientation of the normal of each air-water interface with respect to the direction of the electron beam, and thus the tilt of the particles local to each air-water interface. We have found that air-water interfaces are tilted between 0° and 16° relative to the electron beam when at a nominal stage tilt of 0° (*Table 2*). The average tilt ± (one standard deviation and measurement error) of particle layers at the centers of holes is 4.8°±3.1° (N = 89) and at the edges of holes is 6.9°±3.5° (N = 61) (*Table 2*, *Figure 3*, *Figure 4*). These tilts may be due to a combination of errors in stage orientation, local grid deformations, and/or local air-water interface curvatures. In most cases, these tilts are not systematic with

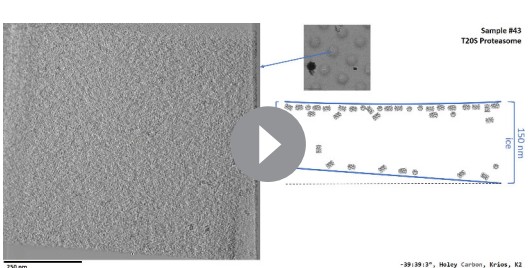

**Video 29.** Sample 44.
DOI: https://doi.org/10.7554/eLife.34257.042

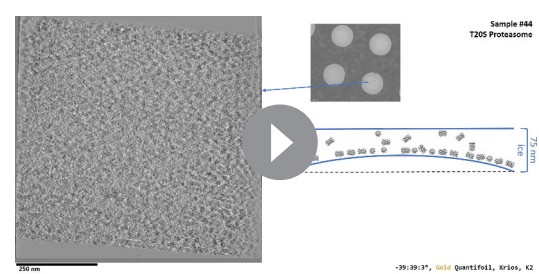

**Video 30.** Sample 45.
DOI: https://doi.org/10.7554/eLife.34257.043

respect to particle orientation in the ice, and thus contribute beneficially to angular particle coverage.

As shown previously, most particles are adsorbed to an air-water interface (*Tables 1* and *2*, *Figure 3*, *Figure 4*, and Videos). It is important to note that a lack of apparent preferred orientation in single particle micrographs does not imply that the particles are not adsorbed to the air-water interfaces. Indeed, most of the particles listed in *Tables 1* and *2* that have no apparent preferred orientations are adsorbed to the air-water interfaces. *Figure 5* shows a selection of adsorbed particles with and without preferred orientations. A distinction should be made between preferred orientation and apparent preferred orientation of particles. A particle may have N and/or M preferred orientations on the grid as shown in *Figure 2B*. Collection on a given grid with non-zero tilts effectively increases the number of imaged preferred orientations of the particle. Depending on the numbers N and/or M, the locations of the preferred orientations on the particle, the symmetry of the particle, and the range of non-zero tilts on the grid, a preferentially oriented particle might have no apparent preferred orientations in a full single particle dataset. As a hypothetical example, both T20S proteasome and apoferritin might have two preferred orientations each, yet T20S proteasome may appear to have a small number of preferred orientations while apoferritin may appear to have no preferred orientations when micrographs are collected with a nominal tilt of zero degrees, but with non-zero degree local air-water interface tilts. This would be due to apoferritin having a high number of uniformly distributed asymmetric units and ~6° tilts in the exposure areas.

The potential effect of tilted particle layers on CTF estimation, and thus resolution limit, of a single particle cryoEM dataset can be nearly as harmful as there being a layer of particles at each air-water interface, as described in the previous section and depicted in *Figure 7*. *Figure 7B* depicts the additional effects imposed by air-water interface and thus particle layer tilts. CTF correction on individual particles using defocus estimation on whole fields of view will limit the resolution of particles above and below the corrected defocus (*Figure 7B*, left and middle) and will alleviate the resolution limit of some particles in thicker areas (*Figure 7B*, middle and right). Additionally, areas of thick ice that are tilted might change which particles are uniquely identifiable (*Figure 7*, right) relative to being untilted. As a hypothetical example, consider a micrograph with a single particle layer in the exposure area and a particle layer tilt of 10° collected at 1 Å pixelsize on a 4 k × 4 k camera with a defocus range of 1 to 2 microns. If the CTF for this micrograph is estimated and corrected for on a whole-image basis, then the worst-corrected particles will have a resolution limit of around 4 Å. These particles might be down-weighted or removed during processing, effectively decreasing the efficiency of the collection.

Several datasets in *Tables 1* and *2* exhibit both of the issues described in this section and in the previous section: overlapping particles in the direction of the electron beam and tilted exposure areas (*Figure 4*, *Figure 7B*). Most of these locations are near hole edges where the ice is often curved and thicker. It is not uncommon for a user to collect single particle micrographs near the edges of holes in order to maximize the collection area in each hole, to avoid the potentially greater beam-induced motion in the center of the holes, and/or to avoid the thin center of holes that are more prone to tearing during exposure. Without previously characterizing the sample in the grid holes by cryoET, collection in these areas might severely limit the number of alignable particles due to projection overlap, the resolution due to CTF estimation and correction error, and the signal due to ice thickness. Thus, for many samples, it is advisable to first determine the distance from the edge of a representative grid hole to collect in order to reliably image single layered particles in thin ice. Doing so will increase the signal due to ice thickness and the reliability and efficiency of single particle alignment and classification due to there being no overlapping particles. CTF estimation and correction should also be performed with the assumption that the field of view is tilted relative to the electron beam (see *Figure 3*), either by performing estimation and correction with whole-image CTF tilt processing, local CTF processing, or per-particle CTF processing (*Grigorieff et al., 2018*; *Hu, 2018*; *Zhang, 2016*). If the ice in thinner areas in the centers of holes is prone to tearing, then one solution might be to image at a lower dose rate.

## Fiducial-less cryoET may be used to determine optimal single particle collection areas and strategies

As shown in *Table 1* and *Figure 3*, ice thickness in holes is commonly greater at the edges than in the centers. Most samples that have this ice behavior have a single layer of particles on one air-water

interface, with either a second layer on the apposed air-water interface or additional non-adsorbed particles, or both (*Figure 3*). At a certain distance from the edge of the holes (usually between 100 to 500 nm from the edge), the ice commonly becomes thin enough for only one layer of particles to fit between – usually the particle's minor axis plus 10 to 20 nm of space between the particles and the air-water interfaces. Provided that particle concentration is high enough for accurate CTF estimation, specimen drift is low enough for sufficient correction, and the particles have little or no apparent preferred orientation, then collection a certain distance away from the edges of these holes would be the most efficient use of resources. Collection in these areas would be less likely to result in anomalous structures compared with collecting in thicker areas with overlapping particles in projections (*Figure 8*, left). If in the same case the

particles show preferred orientations in tomography, then the second most efficient and accurate collection method would be collecting while intentionally tilting the stage (*Tan et al., 2017*), provided that the sample drift is sufficiently low and the concentration is not so high that neighboring particles begin to overlap in the tilted projections (*Figure 8*, middle).

However, if the ice is consistently thick across the holes and across the grid, and/or there is a significant number of overlapping particles in the direction of the electron beam, then it might be determined from cryoET that the sample is not fit for high resolution collection (*Figure 8*, right). If the type of grid used is lacey, then tomography at hole magnification where the imaging area includes several hole sizes may be used to determine hole sizes with thinner ice and to determine if there are one or two particle layers in these areas (*Video 1* for sample #20, deposition data for sample #36). Routinely performing cryoET on cryoEM grids allows for sample owners to determine where and how to collect optimal data most efficiently, or to determine whether or not the grid is collectible to the desired resolution. It takes about 30 to 45 min to collect, process, and analyze a single tomogram. Thus, routine single particle grid and sample characterization by cryoET may not only provide information for optimizing grid preparation of a particular sample, but may also increase microscope efficiency.

## Fiducial-less cryoET may be used to understand critical protein behavior

During the course of this study, cryoET of single particle cryoEM grids has been valuable and even critical for understanding particle stoichiometry and anomalous behavior. For example, cryoET has been used on several HIV-1 trimer preparations with receptors to understand the stoichiometry of the bound receptors by direct visualization of individual particles in 3D (samples #5–7 corresponding to *Videos 8*, *13* and *14*). In another example, sample #17, the size of the 'glycoprotein with bound lipids' particles varied discretely with the radial distance from the edge of holes (*Figure 4* and *Video 15*). In single particle cryoEM micrographs, this observation was not immediately explicable and would have required a single particle data collection followed by alignment and classification before reliable conclusions could be made. Instead, a single tomogram of the sample was collected and it was observed that near the edges of the hole the particles with lipids existed in two layers at the air-water interfaces. Beyond a radial distance from the edge of about 300 nm where the ice became about 15 nm thin the particles and lipids dissociated, with the particles remaining in a single layer (see *Video 1* for sample #20). A solution to this issue was found where glycosylated particles were prepared using Spotiton with conditions that intentionally created thick ice (*Figure 4*, sample #18). A further example highlighting the importance of using cryoET to understand the behavior of samples on grids is sample #40 (*Figure 4*). This sample consisted of a very low concentration of particles in solution prepared with a carbon layer over holes to increase the concentration in holes. CryoET showed that the particles were forming two layers on the carbon: a layer directly on the carbon with about 60% saturation and a layer scattered on top of the first layer with about 30% saturation. This observation made clear that particle overlap would be an issue in single particle processing and introduced the possibility that since the particle layers were directly touching that this might induce conformational change in some of the particles. Similarly for sample #41 (*Figure 4*), cryoET on particles and DNA strands prepared with carbon over holes revealed that a considerable fraction of projection areas consisted of overlapping particles due to some non-adsorbed particles attached to DNA strands. In this situation, it was determined that single particle cryoEM on this sample would be highly inefficient for studying the complex of interest. In the cases described here, cryoET was an expedient and sometimes indispensable method for determining particle behavior.

## Fiducial-less SPT can generate de novo initial models with no additional preparation

A useful and sometimes critical benefit of being able to perform fiducial-less cryoET on a single particle grid is that the resulting tomograms can be processed through single particle tomography alignment and classification in order to generate de novo templates for single particle micrograph picking and for use as initial models in single particle alignment (*Cong and Ludtke, 2010*). Inconsistencies in *ab initio* reconstructions can lead to structural uncertainties during refinement, as shown in the literature (*Ludtke et al., 2011*). In one example reported here (sample #33 and the corresponding *Video 16*), Gaussian particle picking and 2D classification of DnaB helicase-helicase loader particles from single particle micrographs showed one predominant orientation with apparent C6 symmetry and very few different orientations (*Figure 9A*). Efforts to generate an *ab initio* reconstruction with common-lines approaches (*Elmlund and Elmlund, 2012*; *Ludtke et al., 1999*) failed (*Figure 9A*). We suspected that a reliable template could not be generated due to missing many low contrast side-views and more complete particle picking could not be performed without a reliable template – a classic catch-22.

To ameliorate this problem, five tilt-series in representative areas were collected at the end of a single particle collection session, aligned in Appion-Protomo (*Noble and Stagg, 2015*; *Winkler and Taylor, 2006*), and about 1000 particles were processed through sub-tomogram alignment, classification, and multireference alignment using Dynamo (*Castaño-Díez et al., 2012*; *Castaño-Díez et al., 2017*). This resulted in three de novo initial models, each showing an asymmetric cracked ring (*Figure 9B*), contradicting the C6-symmetric reconstruction determined by 2D classification and common-lines approaches. The most populated class from single particle tomography (SPT) was then used to both template pick the single particle micrographs in Relion (*Scheres, 2012*) and as initial models for single particle alignment, resulting in a 4.1 Å structure of the DnaB helicase-helicase loader (manuscript in preparation) (*Figure 9B*). In this example, cryoET revealed that the apparent symmetry in the prevalent top view particles as seen in the Gaussian picked 2D class averages was in fact a projection of the globally asymmetric particle. There are two key benefits to performing fiducial-less cryoET to generate de novo initial models as opposed to fiducial-based cryoET: (1) No additional gold bead + sample preparation and optimization is involved as with conventional fiducial-based tilt-series alignment and (2) the exact sample from which single particle micrographs are collected is used, thus removing the possibility of sample variation across grid preparations.

## Conclusion

We have shown that over a wide range of single particle cryoEM samples, particle and ice behaviors vary widely, yet the vast majority of particles on grids prepared using conventional techniques and using Spotiton with nanowire grids end up adsorbed to air-water interfaces. This varied behavior shown in *Tables 1* and *2* – varied in particle denaturation, particle preferred orientation, particle overlap in the direction of the beam, particle layer tilt, ice thickness, and ice thickness variation across holes – provides impetus for researchers to routinely perform cryoET on their single particle cryoEM grids. Routine characterization of cryoEM grids allows for the determination of particle behavior, whether a single particle sample might produce desirable results, and optimal collection areas and strategies, thus increasing microscope and single particle processing efficiency. Moreover, cryoET on single particle cryoEM grids can be used to generate de novo initial models through single particle sub-tomogram alignment and classification.

The observation that the vast majority of particles are adsorbed to air-water interfaces warrants further research into methods for avoiding the air-water interface. Possible methods include preparing grids with non-ionic surfactants, using affinity grids, encapsulating particles in carbon layers, encapsulating particles in scaffolds, and, perhaps, faster plunging technologies to outrun air-water interface adsorption. Adding surfactants to single particle sample/grid preparation prior to freezing in order to protect bulk proteins from the air-water interfaces has been proposed and used (*Frederik et al., 1989*), yet might be revisited by adding non-ionic surfactants below the CMC. Alternatively, spreading a layer of surfactant (ionic or non-ionic) onto the surface of the air-water interfaces during grid preparation might both reduce the surfactant-protein interaction in solution along with competitive adsorption, and increase the mechanical strength of the resulting surfactant layer on the air-water interface (*Morris and Gunning, 2008*) (perhaps using a method similar to that

described in [*Vos et al., 2008*]). Affinity substrates, such as carbon, streptavidin, or ionic lipid mono-layers over holes may be used in an attempt to escape the air-water interfaces, and potentially have the additional benefit of requiring lower protein concentrations in solution. However, the usage of affinity grids requires further grid optimization with regard to collecting only in areas where the ice is thick enough to more than cover the particles adsorbed to the affinity substrate, and signal is degraded due to the affinity substrate. Encapsulating two-dimensional crystals between carbon layers in order to avoid excessive dehydration due to open air-water interfaces has been performed successfully (*Yang et al., 2013*), opening up the possibility of encapsulating particles in-between car-bon, or possibly graphene layers, to avoid air-water interface interactions. Particle encapsulation using protein scaffolds (*Kedersha and Rome, 1986*) or synthetic DNA structures (*Martin et al., 2016*) has also been proposed for avoiding air-water interface and preferred orientation issues. Lastly, decreasing the time between sample application and freezing in order to outrun air-water interface adsorption altogether might be possible with further technological development (*Arnold et al., 2017*; *Feng et al., 2017*; *Frank, 2017*; *Jain et al., 2012*; *Noble et al., 2018*). The time it takes for a particle to diffuse to an air-water interface, to diffuse across the air-water inter-face, and for subsequent bulk particles to adsorb to the resulting viscoelastic protein network film might be on the order of tens of milliseconds or greater. This process appears to be largely depen-dent on protein surface hydrophobicity, protein concentration, and protein structure. Avoiding the air-water interface may prove critical for obtaining higher resolution structures of more fragile proteins.

## Materials and methods

### Grid preparation

About one-third of the grids characterized were prepared using conventional techniques as deter-mined by the sample owner. Generally, a purchased holey grid (most were Quantifoil (Quantifoil Micro Tools, GmbH, Jena, Germany) or C-flat (Protochips, Inc., Morrisville, North Carolina) carbon or gold) was glow-discharged, sample was applied at appropriate conditions, incubation on the order of 1 to 10 s took place, the grid was blotted (most commonly face blotted), further incubation on the order of 1 s took place, and then the grid was plunged into liquid ethane.

The remaining grids were prepared using Spotiton (*Jain et al., 2012*). Generally, a home-made lacey or holey carbon or gold nanowire grid (*Razinkov et al., 2016*) was glow-discharged, sample was sprayed onto the grid in a stripe, incubation on the order of 1 s or less took place as determined by the calibrated self-wicking time or by the maximum plunging speed of the robot, and then the grid was plunged into liquid ethane.

### Tilt-series collection

Tilt-series were collected at NYSBC on one of the Titan Krios microscopes (FEI Company, Hillsboro, OR) with a Gatan K2 (Gatan, Inc., Pleasanton, CA) or on the Tecnai F20 (FEI Company, Hillsboro, OR) with a DE-20 (Direct Electron, San Diego, CA) or a Tietz F416 (TVIPS GmbH, Gauting, Germany). Several tilt-series were collected using a Gatan Bioquantum energy filter (Gatan, Inc., Pleasanton, CA), and a small number were collected with a Volta phase plate (FEI Company, Hillsboro, OR). Most tilt-series were collected using Leginon (*Suloway et al., 2005*; *Suloway et al., 2009*) on the Krios microscopes and the F20, with the remaining collected using SerialEM (*Mastronarde, 2003*) on the F20. Most tilt-series were collected with 100 ms frames for each tilt image and full-frame aligned using MotionCor2 (*Zheng et al., 2017*). Most tilt-series were collected bi-directionally with a tilt range of −45° to 45° and a tilt increment of 3°. Most tilt-series were collected at a nominal defo-cus between 4 to 6 microns. Most tilt-series were collected with a dose rate around 8 e-/pixel/s and an incident dose between 1.5 and 3.0 e-/Å$^2$ for the zero-degree tilt image, with increasing dose for higher tilt angles according to the cosine of the tilt angle, resulting in a total dose between 50 and 150 e-/Å$^2$. Most tilt-series were collected at a pixelsize between 1 and 2.2 Å. Hole magnification tilt-series were typically collected with a tilt range of −60° to 60° with a tilt increment of 1°, a pixelsize around 20 Å, and negligible dose. Each high-magnification tilt-series typically collect in around 15 min, while hole magnification tilt-series take about 30 min. Most tilt-series were collected without hardware binning. Two samples were collected using super-resolution.

## Tilt-series alignment

Tilt-series collected with Leginon are automatically available for processing in Appion (*Lander et al., 2009*), while tilt-series collected with SerialEM (*Mastronarde, 2003*) were uploaded to Appion prior to alignment. All tilt-series were aligned using Appion-Protomo (*Noble and Stagg, 2015*). Briefly, most tilt-series were first dose compensated using the relation in (*Grant and Grigorieff, 2015*), coarsely aligned, manually aligned if necessary, refined using a set of alignment thicknesses, then the best aligned iteration was reconstructed for visual analysis using Tomo3D SIRT (*Agulleiro and Fernandez, 2011*; *Agulleiro and Fernandez, 2015*). CTF correction was not performed. Tilt-series typically align well in 20–60 min. Nearly all tilt-series were alignable.

## CTF resolution limit

Resolution limits due to errors in defocus estimation as reported in the Results and Discussion were determined by plotting two CTF curves at about 1.5 microns defocus but differing by defocus error and locating the approximate resolution where the curves are out of phase by 90°.

## Estimations and measurement error

Ice thickness measurements were performed as follows: After orienting a binned by four high magnification tomogram (pixel size of about 8 Å) or an unbinned hole magnification tomogram (pixel size of about 20 Å) in 3dmod such the one air-water interface is approximately parallel to the field of view, either contamination local to the surface of the ice or an adsorbed particle layer was used to locate the two air-water interfaces, and the distance between the two interfaces was measured. If contamination was used, then the tomogram slice nearest to the vitreous ice and still containing the contamination was used to locate the interface. If particles were used, then then the tomogram slice nearest to the air and still containing the particles was used to locate the interface. For these measurements, the estimated error in measuring ice thickness and particle layer distance from the air-water interface is several nanometers for high magnification and ~10 nm for hole magnification.

Statistical and systematic errors for measurements presented in *Figure 3* were propagated as follows. Each reported value for ice thickness and particle layer tilt is reported with an estimated error that is the sum under the quadrature of the standard deviation and the propagated measurement error. The standard deviation was calculated using all measured values (indicated by N size). For measurement error, ice thickness measurements contain an approximate error of 5 nm for each measurement and particle layer tilt contain an approximate error of 1°. Measurement error of the average values presented in *Figure 3* was propagated by assuming independent random errors using the following equation:

$$\delta q = \frac{\sqrt{\sum((\delta x)^2)}}{N}$$

where $\delta q$ is the propagated measurement error, $\delta x$ is each independent measurement error, and N is the sample size. Most propagated measurement errors are an order of magnitude less than the standard deviation.

The smoothness of the depicted ice surfaces is an approximation.

## Data deposition and software availability

Several representative tilt-series from the datasets have been deposited to the Electron Microscopy Data Bank (EMDB) in the form of binned by 4 or 8 tomograms and to the Electron Microscopy Pilot Image Archive (EMPIAR) in the form of unaligned tilt-series images (one including super-resolution frames), Appion-Protomo tilt-series alignment runs, and aligned tilt-series stacks. Their accession codes are:

| Sample # | Sample name | EMDB (tomogram) | EMPIAR (tomogram) | EMPIAR (single particle) |
|---|---|---|---|---|
| 4 | Hemagglutinin | 7135 | 10129 | – |

*Continued on next page*

*Continued*

| Sample # | Sample name | EMDB (tomogram) | EMPIAR (tomogram) | EMPIAR (single particle) |
|---|---|---|---|---|
| 21 | Rabbit Muscle Aldolase (1 mg/mL) | 7138 | 10130 | – |
| 22 | Rabbit Muscle Aldolase (6 mg/mL) | 7139 | 10131 | 10187 |
| 25 | Protein in Nanodisc (0.58 mg/mL) | 7140 | – | – |
| 30 | GDH | 7141 | 10132 | 10132 |
| 31 | GDH | 7142 | 10133 | – |
| 32 | GDH (2.5 mg/mL)+0.001% DDM | 7143 | 10134 | 10134 |
| 33 | DnaB Helicase-helicase Loader | 7144 | 10135 | – |
| 34 | Apoferritin | 7145 | 10136 | – |
| 35 | Apoferritin | 7146 | 10137 | – |
| 36 | Apoferritin | 7147 | 10138 | 10138 |
| 37 | Apoferritin (1.25 mg/mL) | 7148 | 10139 | – |
| 38 | Apoferritin (0.5 mg/mL) | 7149 | 10140 | – |
| 39 | Apoferritin with 0.5 mM TCEP | 7150 | 10141 | – |
| 42 | T20S Proteasome | 7151 | 10142 | – |
| 43 | T20S Proteasome | 7152 | 10143 | 10143 |
| 44 | T20S Proteasome | 7153 | 10144 | 10188 |
| 45 | Mtb 20S Proteasome | 7154 | 10145 | – |

Protomo estimations for the orientation of the local ice normal based on the tilt-series alignment of the particles in the ice, which includes potential systematic stage and beam axis error, are available in all deposited EMPIAR datasets as a plot located: protomo_alignments/tiltseries####/media/angle_refinement/series####_orientation.gif

A Docker-based version of Appion-Protomo fiducial-less tilt-series alignment is available at https://github.com/nysbc/appion-protomo.

## Videos

Each Video (except for sample #20) shows slice-throughs (with bottom/top oriented as described in the text) of one tomogram from a given sample in *Table 1 and 2* alongside a schematic cross-sectional diagram of the sample and the ice. Most tomograms are oriented such that the plane of one of the particle layers is parallel to the viewing plane. A hole magnification tomogram is shown in the Video for sample #20. The tomograms were rendered with 3dmod from the IMOD package (*Kremer et al., 1996*) and the schematic particles were rendered with UCSF Chimera (*Pettersen et al., 2004*).

## Acknowledgements

The authors wish to thank Prof. Robert Glaeser (Lawrence Berkeley National Laboratory) and Prof. Pete Wilde (Quadram Institute) for helpful discussions. The authors wish to thank Neil Voss (Roosevelt University) for the base Appion CentOS7 Docker image. The authors wish to thank several unnamed sample sources. The Mtb 20S Proteasome sample was a kind gift from Huilin Li (Van Andel Research Institute). Some of this work was performed at the Simons Electron Microscopy Center and National Resource for Automated Molecular Microscopy located at the New York Structural Biology Center, supported by grants from the Simons Foundation (SF349247), NYSTAR, and the NIH National Institute of General Medical Sciences (GM103310) with additional support from the Agouron Institute (F00316) and NIH (OD019994). Studies on stick-like particles and neural receptors were supported in part by the National Institutes of Health (R01-MH1148175). Science in the

Jeruzalmi lab was supported by the National Institutes of Health (R01 GM084162), and the National Institute on Minority Health and Health Disparities (5G12MD007603-30). Studies on the HIV-1 trimers were supported in part by Intramural Funding from the Vaccine Research Center, NIAID, NIH. Studies on the protein in nanodisc were supported in part by the Agency for Science, Technology and Research Singapore.

## Additional information

### Funding

| Funder | Grant reference number | Author |
|---|---|---|
| Simons Foundation | SF349247 | Clinton S Potter Bridget Carragher |
| New York State Foundation for Science, Technology and Innovation | | Clinton S Potter Bridget Carragher |
| National Institute of General Medical Sciences | GM103310 | Clinton S Potter Bridget Carragher |
| Agouron Institute | F00316 | Clinton S Potter Bridget Carragher |
| National Institutes of Health | S10 OD019994-01 | Clinton S Potter Bridget Carragher |
| National Institute on Minority Health and Health Disparities | 5G12MD007603-30 | David Jeruzalmi |
| National Institute of Allergy and Infectious Diseases | Intramural Funding from the Vaccine Research Center | Peter D Kwong |
| Agency for Science, Technology and Research | | Yong Zi Tan |
| National Institutes of Health | R01-MH1148175 | Lawrence Shapiro |
| National Institutes of Health | R01 GM084162 | David Jeruzalmi |

The authors declare that the funders played no role in this work, including the experimental design, data collection, or data analysis.

### Author contributions

Alex J Noble, Conceptualization, Data curation, Software, Formal analysis, Validation, Investigation, Visualization, Methodology, Writing—original draft, Project administration, Writing—review and editing; Venkata P Dandey, Hui Wei, Zhening Zhang, Laura Y Kim, Giovanna Scapin, Micah Rapp, William J Rice, Investigation; Julia Brasch, Priyamvada Acharya, Yong Zi Tan, Edward T Eng, Investigation, Writing—review and editing; Jillian Chase, Formal analysis, Investigation, Visualization, Writing—review and editing; Anchi Cheng, Software, Investigation; Carl J Negro, Software; Lawrence Shapiro, Peter D Kwong, David Jeruzalmi, Resources, Supervision, Funding acquisition; Amedee des Georges, Resources, Supervision, Funding acquisition, Writing—review and editing; Clinton S Potter, Bridget Carragher, Conceptualization, Resources, Supervision, Funding acquisition, Project administration, Writing—review and editing

### Author ORCIDs

Alex J Noble http://orcid.org/0000-0001-8634-2279

Yong Zi Tan http://orcid.org/0000-0001-6656-6320

Edward T Eng http://orcid.org/0000-0002-8014-7269

David Jeruzalmi http://orcid.org/0000-0001-5886-1370

Clinton S Potter http://orcid.org/0000-0003-2394-0831

Bridget Carragher http://orcid.org/0000-0002-0624-5020

Decision letter and Author response
Decision letter https://doi.org/10.7554/eLife.34257.120
Author response https://doi.org/10.7554/eLife.34257.121

# Additional files

## Supplementary files

• Transparent reporting form
DOI: https://doi.org/10.7554/eLife.34257.044

## Data availability

Several representative tilt-series from the datasets have been deposited to the Electron Microscopy Data Bank (EMDB) in the form of binned by 4 or 8 tomograms and to the Electron Microscopy Pilot Image Archive (EMPIAR) in the form of unaligned tilt-series images (one including super-resolution frames), Appion-Protomo tilt-series alignment runs, and aligned tilt-series stacks. Protomo estimations for the orientation of the local ice normal based on the tilt-series alignment of the particles in the ice, which includes potential systematic stage and beam axis error, are available in all deposited EMPIAR datasets as a plot located: protomo_alignments/tiltseries####/media/angle_refinement/series####_orientation.gif A Docker-based version of Appion-Protomo fiducial-less tilt-series alignment is available at http://github.com/nysbc/appion-protomo.

The following datasets were generated:

| Author(s) | Year | Dataset title | Dataset URL | Database, license, and accessibility information |
|---|---|---|---|---|
| Alex J Noble, Venkata P Dandey, Hui Wei, Julia Brasch, Jillian Chase, Priyamvada Acharya, Yong Zi Tan, Zhening Zhang, Laura Y Kim, Giovanna Scapin, Micah Rapp, Edward T Eng, William J Rice, Anchi Cheng, Carl J Negro, Lawrence Shapiro, Peter D Kwong, David Jeruzalmi, Amedee des Georges, Clinton S Potter, Bridget Carragher | 2018 | Rabbit muscle aldolase single particle | https://www.ebi.ac.uk/pdbe/emdb/empiar/entry/10187 | Publicly available at the Electron Microscopy Data Bank (accession no. EMPIAR-10187) |
| Alex J Noble, Venkata P Dandey, Hui Wei, Julia Brasch, Jillian Chase, Priyamvada Acharya, Yong Zi Tan, Zhening Zhang, Laura Y Kim, Giovanna Scapin, Micah Rapp, Edward T Eng, William J Rice, Anchi Cheng, Carl J Negro, Lawrence Shapiro, Peter D Kwong, David Jeruzalmi, Amedee des Georges, Clinton S Potter, Bridget Carragher | 2017 | Mtb 20S proteasome on a carbon nanowire grid plunged with Spotiton | https://www.ebi.ac.uk/pdbe/entry/emdb/EMD-7154 | Publicly available at the Electron Microscopy Data Bank (accession no. EMD-7154) |
| Alex J Noble, Venkata P Dandey, Hui | 2018 | T20S proteasome single particle | https://www.ebi.ac.uk/pdbe/emdb/empiar/en- | Publicly available at the Electron |

| | | | | | |
|---|---|---|---|---|---|
| Wei, Julia Brasch, Jillian Chase, Priyamvada Acharya, Yong Zi Tan, Zhening Zhang, Laura Y Kim, Giovanna Scapin, Micah Rapp, Edward T Eng, William J Rice, Anchi Cheng, Carl J Negro, Lawrence Shapiro, Peter D Kwong, David Jeruzalmi, Amedee des Georges, Clinton S Potter, Bridget Carragher | | | try/10188 | Microscopy Data Bank (accession no. EMPIAR-10188) | |
| Noble AJ, Dandey VP, Wei H, Brasch J, Chase J, Acharya P, Tan YZ, Zhang Z, Kim LY, Scapin G, Rapp M, Eng ET, Rice WJ, Cheng A, Negro CJ, Shapiro L, Kwong PD, Jeruzalmi D, des Georges A, Potter CS, Carragher B | 2017 | CryoET of hemagglutinin single particle | https://www.ebi.ac.uk/pdbe/emdb/empiar/entry/10129 | Publicly available at the Electron Microscopy Data Bank (accession no. EMPIAR-10129) | |
| Noble AJ, Dandey VP, Wei H, Brasch J, Chase J, Acharya P, Tan YZ, Zhang Z, Kim LY, Scapin G, Rapp M, Eng ET, Rice WJ, Cheng A, Negro CJ, Shapiro L, Kwong PD, Jeruzalmi D, des Georges A, Potter CS, Carragher B | 2017 | CryoET of rabbit muscle aldolase single particle super-res | https://www.ebi.ac.uk/pdbe/emdb/empiar/entry/10130 | Publicly available at the Electron Microscopy Data Bank (accession no. EMPIAR-10130) | |
| Noble AJ, Dandey VP, Wei H, Brasch J, Chase J, Acharya P, Tan YZ, Zhang Z, Kim LY, Scapin G, Rapp M, Eng ET, Rice WJ, Cheng A, Negro CJ, Shapiro L, Kwong PD, Jeruzalmi D, des Georges A, Potter CS, Carragher B | 2017 | CryoET of rabbit muscle aldolase single particle | https://www.ebi.ac.uk/pdbe/emdb/empiar/entry/10131 | Publicly available at the Electron Microscopy Data Bank (accession no. EMPIAR-10131) | |
| Noble AJ, Dandey VP, Wei H, Brasch J, Chase J, Acharya P, Tan YZ, Zhang Z, Kim LY, Scapin G, Rapp M, Eng ET, Rice WJ, Cheng A, Negro CJ, Shapiro L, Kwong PD, Jeruzalmi D, des Georges A, Potter CS, Carragher B | 2017 | CryoET of glutamate dehydrogenase single particle | https://www.ebi.ac.uk/pdbe/emdb/empiar/entry/10132 | Publicly available at the Electron Microscopy Data Bank (accession no. EMPIAR-10132) | |
| Noble AJ, Dandey VP, Wei H, Brasch J, Chase J, Acharya P, Tan YZ, Zhang Z, Kim LY, Scapin G, Rapp M, Eng ET, | 2017 | CryoET of glutamate dehydrogenase single particle | https://www.ebi.ac.uk/pdbe/emdb/empiar/entry/10133 | Publicly available at the Electron Microscopy Data Bank (accession no. EMPIAR-10133) | |

| | | | | | |
|---|---|---|---|---|---|
| Rice WJ, Cheng A, Negro CJ, Shapiro L, Kwong PD, Jeruzalmi D, des Georges A, Potter CS, Carragher B | | | | | |
| Noble AJ, Dandey VP, Wei H, Brasch J, Chase J, Acharya P, Tan YZ, Zhang Z, Kim LY, Scapin G, Rapp M, Eng ET, Rice WJ, Cheng A, Negro CJ, Shapiro L, Kwong PD, Jeruzalmi D, des Georges A, Potter CS, Carragher B | 2017 | CryoET of glutamate dehydrogenase + 0.001% DDM single particle | https://www.ebi.ac.uk/pdbe/emdb/empiar/entry/10134 | | Publicly available at the Electron Microscopy Data Bank (accession no. EMPIAR-10134) |
| Noble AJ, Dandey VP, Wei H, Brasch J, Chase J, Acharya P, Tan YZ, Zhang Z, Kim LY, Scapin G, Rapp M, Eng ET, Rice WJ, Cheng A, Negro CJ, Shapiro L, Kwong PD, Jeruzalmi D, des Georges A, Potter CS, Carragher B | 2017 | CryoET of DNAB helicase-helicase loader single particle | https://www.ebi.ac.uk/pdbe/emdb/empiar/entry/10135 | | Publicly available at the Electron Microscopy Data Bank (accession no. EMPIAR-10135) |
| Noble AJ, Dandey VP, Wei H, Brasch J, Chase J, Acharya P, Tan YZ, Zhang Z, Kim LY, Scapin G, Rapp M, Eng ET, Rice WJ, Cheng A, Negro CJ, Shapiro L, Kwong PD, Jeruzalmi D, des Georges A, Potter CS, Carragher B | 2017 | CryoET of apoferritin single particle | https://www.ebi.ac.uk/pdbe/emdb/empiar/entry/10136 | | Publicly available at the Electron Microscopy Data Bank (accession no. EMPIAR-10136) |
| Noble AJ, Dandey VP, Wei H, Brasch J, Chase J, Acharya P, Tan YZ, Zhang Z, Kim LY, Scapin G, Rapp M, Eng ET, Rice WJ, Cheng A, Negro CJ, Shapiro L, Kwong PD, Jeruzalmi D, des Georges A, Potter CS, Carragher B | 2017 | CryoET of apoferritin single particle | https://www.ebi.ac.uk/pdbe/emdb/empiar/entry/10137 | | Publicly available at the Electron Microscopy Data Bank (accession no. EMPIAR-10137) |
| Noble AJ, Dandey VP, Wei H, Brasch J, Chase J, Acharya P, Tan YZ, Zhang Z, Kim LY, Scapin G, Rapp M, Eng ET, Rice WJ, Cheng A, Negro CJ, Shapiro L, Kwong PD, Jeruzalmi D, des Georges A, Potter CS, Carragher B | 2017 | CryoET of apoferritin single particle | https://www.ebi.ac.uk/pdbe/emdb/empiar/entry/10138 | | Publicly available at the Electron Microscopy Data Bank (accession no. EMPIAR-10138) |
| Noble AJ, Dandey VP, Wei H, Brasch J, Chase J, Acharya P, Tan YZ, Zhang Z, Kim LY, Scapin G, | 2017 | CryoET of apoferritin single particle | https://www.ebi.ac.uk/pdbe/emdb/empiar/entry/10139 | | Publicly available at the Electron Microscopy Data Bank (accession no. EMPIAR-10139) |

| | | | | |
|---|---|---|---|---|
| Rapp M, Eng ET, Rice WJ, Cheng A, Negro CJ, Shapiro L, Kwong PD, Jeruzalmi D, des Georges A, Potter CS, Carragher B | | | | |
| Noble AJ, Dandey VP, Wei H, Brasch J, Chase J, Acharya P, Tan YZ, Zhang Z, Kim LY, Scapin G, Rapp M, Eng ET, Rice WJ, Cheng A, Negro CJ, Shapiro L, Kwong PD, Jeruzalmi D, des Georges A, Potter CS, Carragher B | 2017 | CryoET of apoferritin single particle | https://www.ebi.ac.uk/pdbe/emdb/empiar/entry/10140 | Publicly available at the Electron Microscopy Data Bank (accession no. EMPIAR-10140) |
| Noble AJ, Dandey VP, Wei H, Brasch J, Chase J, Acharya P, Tan YZ, Zhang Z, Kim LY, Scapin G, Rapp M, Eng ET, Rice WJ, Cheng A, Negro CJ, Shapiro L, Kwong PD, Jeruzalmi D, des Georges A, Potter CS, Carragher B | 2017 | CryoET of apoferritin with 0.5 mM TCEP single particle | https://www.ebi.ac.uk/pdbe/emdb/empiar/entry/10141 | Publicly available at the Electron Microscopy Data Bank (accession no. EMPIAR-10141) |
| Noble AJ, Dandey VP, Wei H, Brasch J, Chase J, Acharya P, Tan YZ, Zhang Z, Kim LY, Scapin G, Rapp M, Eng ET, Rice WJ, Cheng A, Negro CJ, Shapiro L, Kwong PD, Jeruzalmi D, des Georges A, Potter CS, Carragher B | 2017 | Phase plate cryoET of T20S proteasome single particle | https://www.ebi.ac.uk/pdbe/emdb/empiar/entry/10142 | Publicly available at the Electron Microscopy Data Bank (accession no. EMPIAR-10142) |
| Noble AJ, Dandey VP, Wei H, Brasch J, Chase J, Acharya P, Tan YZ, Zhang Z, Kim LY, Scapin G, Rapp M, Eng ET, Rice WJ, Cheng A, Negro CJ, Shapiro L, Kwong PD, Jeruzalmi D, des Georges A, Potter CS, Carragher B | 2017 | CryoET of T20S proteasome single particle | https://www.ebi.ac.uk/pdbe/emdb/empiar/entry/10143 | Publicly available at the Electron Microscopy Data Bank (accession no. EMPIAR-10143) |
| Noble AJ, Dandey VP, Wei H, Brasch J, Chase J, Acharya P, Tan YZ, Zhang Z, Kim LY, Scapin G, Rapp M, Eng ET, Rice WJ, Cheng A, Negro CJ, Shapiro L, Kwong PD, Jeruzalmi D, des Georges A, Potter CS, Carragher B | 2017 | CryoET of T20S proteasome single particle | https://www.ebi.ac.uk/pdbe/emdb/empiar/entry/10144 | Publicly available at the Electron Microscopy Data Bank (accession no. EMPIAR-10144) |
| Noble AJ, Dandey VP, Wei H, Brasch J, Chase J, Acharya P, Tan YZ, Zhang Z, | 2017 | CryoET of Mtb 20S proteasome single particle | https://www.ebi.ac.uk/pdbe/emdb/empiar/entry/10145 | Publicly available at the Electron Microscopy Data Bank (accession no. |

| | | | | |
|---|---|---|---|---|
| Kim LY, Scapin G, Rapp M, Eng ET, Rice WJ, Cheng A, Negro CJ, Shapiro L, Kwong PD, Jeruzalmi D, des Georges A, Potter CS, Carragher B | | | | EMPIAR-10145) |
| Noble AJ, Dandey VP, Wei H, Brasch J, Chase J, Acharya P, Tan YZ, Zhang Z, Kim LY, Scapin G, Rapp M, Eng ET, Rice WJ, Cheng A, Negro CJ, Shapiro L, Kwong PD, Jeruzalmi D, des Georges A, Potter CS, Carragher B | 2017 | Hemagglutinin on a carbon nanowire grid plunged with Spotiton | https://www.ebi.ac.uk/pdbe/entry/emdb/EMD-7135 | Publicly available at the Electron Microscopy Data Bank (accession no. EMD-7135) |
| Noble AJ, Dandey VP, Wei H, Brasch J, Chase J, Acharya P, Tan YZ, Zhang Z, Kim LY, Scapin G, Rapp M, Eng ET, Rice WJ, Cheng A, Negro CJ, Shapiro L, Kwong PD, Jeruzalmi D, des Georges A, Potter CS, Carragher B | 2017 | Rabbit muscle aldolase on a gold nanowire grid plunged with Spotiton | https://www.ebi.ac.uk/pdbe/entry/emdb/EMD-7138 | Publicly available at the Electron Microscopy Data Bank (accession no. EMD-7138) |
| Noble AJ, Dandey VP, Wei H, Brasch J, Chase J, Acharya P, Tan YZ, Zhang Z, Kim LY, Scapin G, Rapp M, Eng ET, Rice WJ, Cheng A, Negro CJ, Shapiro L, Kwong PD, Jeruzalmi D, des Georges A, Potter CS, Carragher B | 2017 | Rabbit muscle aldolase on a carbon nanowire grid plunged with Spotiton | https://www.ebi.ac.uk/pdbe/entry/emdb/EMD-7139 | Publicly available at the Electron Microscopy Data Bank (accession no. EMD-7139) |
| Noble AJ, Dandey VP, Wei H, Brasch J, Chase J, Acharya P, Tan YZ, Zhang Z, Kim LY, Scapin G, Rapp M, Eng ET, Rice WJ, Cheng A, Negro CJ, Shapiro L, Kwong PD, Jeruzalmi D, des Georges A, Potter CS, Carragher B | 2017 | Protein in nanodisc on a gold nanowire grid plunged with Spotiton | https://www.ebi.ac.uk/pdbe/entry/emdb/EMD-7140 | Publicly available at the Electron Microscopy Data Bank (accession no. EMD-7140) |
| Noble AJ, Dandey VP, Wei H, Brasch J, Chase J, Acharya P, Tan YZ, Zhang Z, Kim LY, Scapin G, Rapp M, Eng ET, Rice WJ, Cheng A, Negro CJ, Shapiro L, Kwong PD, Jeruzalmi D, des Georges A, Potter CS, Carragher B | 2017 | Glutamate dehydrogenase on a holey carbon grid | https://www.ebi.ac.uk/pdbe/entry/emdb/EMD-7141 | Publicly available at the Electron Microscopy Data Bank (accession no. EMD-7141) |
| Noble AJ, Dandey VP, Wei H, Brasch J, Chase J, Acharya | 2017 | Glutamate dehydrogenase on a holey carbon grid | https://www.ebi.ac.uk/pdbe/entry/emdb/EMD-7142 | Publicly available at the Electron Microscopy Data |

| | | | | |
|---|---|---|---|---|
| P, Tan YZ, Zhang Z, Kim LY, Scapin G, Rapp M, Eng ET, Rice WJ, Cheng A, Negro CJ, Shapiro L, Kwong PD, Jeruzalmi D, des Georges A, Potter CS, Carragher B | | | | Bank (accession no. EMD-7142) |
| Noble AJ, Dandey VP, Wei H, Brasch J, Chase J, Acharya P, Tan YZ, Zhang Z, Kim LY, Scapin G, Rapp M, Eng ET, Rice WJ, Cheng A, Negro CJ, Shapiro L, Kwong PD, Jeruzalmi D, des Georges A, Potter CS, Carragher B | 2017 | GDH + 0.001% DDM on a carbon nanowire grid plunged with Spotiton | https://www.ebi.ac.uk/pdbe/entry/emdb/EMD-7143 | Publicly available at the Electron Microscopy Data Bank (accession no. EMD-7143) |
| Noble AJ, Dandey VP, Wei H, Brasch J, Chase J, Acharya P, Tan YZ, Zhang Z, Kim LY, Scapin G, Rapp M, Eng ET, Rice WJ, Cheng A, Negro CJ, Shapiro L, Kwong PD, Jeruzalmi D, des Georges A, Potter CS, Carragher B | 2017 | DNAB helicase-helicase loader on a gold Quantifoil grid | https://www.ebi.ac.uk/pdbe/entry/emdb/EMD-7144 | Publicly available at the Electron Microscopy Data Bank (accession no. EMD-7144) |
| Noble AJ, Dandey VP, Wei H, Brasch J, Chase J, Acharya P, Tan YZ, Zhang Z, Kim LY, Scapin G, Rapp M, Eng ET, Rice WJ, Cheng A, Negro CJ, Shapiro L, Kwong PD, Jeruzalmi D, des Georges A, Potter CS, Carragher B | 2017 | Apoferritin on a gold nanowire grid plunged with Spotiton | https://www.ebi.ac.uk/pdbe/entry/emdb/EMD-7145 | Publicly available at the Electron Microscopy Data Bank (accession no. EMD-7145) |
| Noble AJ, Dandey VP, Wei H, Brasch J, Chase J, Acharya P, Tan YZ, Zhang Z, Kim LY, Scapin G, Rapp M, Eng ET, Rice WJ, Cheng A, Negro CJ, Shapiro L, Kwong PD, Jeruzalmi D, des Georges A, Potter CS, Carragher B | 2017 | Apoferritin on a gold nanowire grid plunged with Spotiton | https://www.ebi.ac.uk/pdbe/entry/emdb/EMD-7146 | Publicly available at the Electron Microscopy Data Bank (accession no. EMD-7146) |
| Noble AJ, Dandey VP, Wei H, Brasch J, Chase J, Acharya P, Tan YZ, Zhang Z, Kim LY, Scapin G, Rapp M, Eng ET, Rice WJ, Cheng A, Negro CJ, Shapiro L, Kwong PD, Jeruzalmi D, des Georges A, Potter CS, Carragher B | 2017 | Apoferritin on a holey carbon nanowire grid plunged with Spotiton | https://www.ebi.ac.uk/pdbe/entry/emdb/EMD-7147 | Publicly available at the Electron Microscopy Data Bank (accession no. EMD-7147) |
| Noble AJ, Dandey VP, Wei H, Brasch | 2017 | Apoferritin on a holey carbon nanowire grid plunged with | https://www.ebi.ac.uk/pdbe/entry/emdb/EMD- | Publicly available at the Electron |

| | | | | |
|---|---|---|---|---|
| J, Chase J, Acharya P, Tan YZ, Zhang Z, Kim LY, Scapin G, Rapp M, Eng ET, Rice WJ, Cheng A, Negro CJ, Shapiro L, Kwong PD, Jeruzalmi D, des Georges A, Potter CS, Carragher B | | Spotiton | 7148 | Microscopy Data Bank (accession no. EMD-7148) |
| Noble AJ, Dandey VP, Wei H, Brasch J, Chase J, Acharya P, Tan YZ, Zhang Z, Kim LY, Scapin G, Rapp M, Eng ET, Rice WJ, Cheng A, Negro CJ, Shapiro L, Kwong PD, Jeruzalmi D, des Georges A, Potter CS, Carragher B | 2017 | Apoferritin on a holey gold nanowire grid plunged with Spotiton | https://www.ebi.ac.uk/pdbe/entry/emdb/EMD-7149 | Publicly available at the Electron Microscopy Data Bank (accession no. EMD-7149) |
| Noble AJ, Dandey VP, Wei H, Brasch J, Chase J, Acharya P, Tan YZ, Zhang Z, Kim LY, Scapin G, Rapp M, Eng ET, Rice WJ, Cheng A, Negro CJ, Shapiro L, Kwong PD, Jeruzalmi D, des Georges A, Potter CS, Carragher B | 2017 | Apoferritin with 0.5 mM TCEP on a carbon nanowire grid plunged with Spotiton | https://www.ebi.ac.uk/pdbe/entry/emdb/EMD-7150 | Publicly available at the Electron Microscopy Data Bank (accession no. EMD-7150) |
| Noble AJ, Dandey VP, Wei H, Brasch J, Chase J, Acharya P, Tan YZ, Zhang Z, Kim LY, Scapin G, Rapp M, Eng ET, Rice WJ, Cheng A, Negro CJ, Shapiro L, Kwong PD, Jeruzalmi D, des Georges A, Potter CS, Carragher B | 2017 | T20S proteasome on a holey carbon grid | https://www.ebi.ac.uk/pdbe/entry/emdb/EMD-7151 | Publicly available at the Electron Microscopy Data Bank (accession no. EMD-7151) |
| Noble AJ, Dandey VP, Wei H, Brasch J, Chase J, Acharya P, Tan YZ, Zhang Z, Kim LY, Scapin G, Rapp M, Eng ET, Rice WJ, Cheng A, Negro CJ, Shapiro L, Kwong PD, Jeruzalmi D, des Georges A, Potter CS, Carragher B | 2017 | T20S proteasome on a holey carbon grid | https://www.ebi.ac.uk/pdbe/entry/emdb/EMD-7152 | Publicly available at the Electron Microscopy Data Bank (accession no. EMD-7152) |
| Noble AJ, Dandey VP, Wei H, Brasch J, Chase J, Acharya P, Tan YZ, Zhang Z, Kim LY, Scapin G, Rapp M, Eng ET, Rice WJ, Cheng A, Negro CJ, Shapiro L, Kwong PD, Jeruzalmi D, des Georges A, Potter CS, Carragher B | 2017 | T20S proteasome on a gold Quantifoil grid | https://www.ebi.ac.uk/pdbe/entry/emdb/EMD-7153 | Publicly available at the Electron Microscopy Data Bank (accession no. EMD-7153) |

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
