## [Decision Letter]

Thank you for submitting your article "Routine Single Particle CryoEM Sample and Grid Characterization by Tomography" for consideration by *eLife*. Your article has been reviewed by three peer reviewers, and the evaluation has been overseen by a Reviewing Editor and John Kuriyan as the Senior Editor. The following individuals involved in review of your submission have agreed to reveal their identity: Christopher J Russo (Reviewer #1); Georgios Skiniotis (Reviewer #2); John AG Briggs (Reviewer #3).

The reviewers have discussed the reviews with one another and the Reviewing Editor has drafted this decision to help you prepare a revised submission.

Summary:

Noble et al. applied cryo-electron tomography in a large variety of single-particle samples prepared with a range of vitrification devices and grids to characterize the particle distribution in vitreous ice and the ice geometry in vitrified layers. The results demonstrate convincingly that the vast majority of particles in most cases adsorb to the air-water interface, a phenomenon with potential consequences regarding protein denaturation or complex dissociation, non-physiological conformational changes, as well as preferred particle orientation based on the preferential exposure of the most hydrophobic regions. This study, being the first to systematically examine particle localization in vitrified ice, is important given its implications for the rapidly developing cryo-EM field. However, a weakness of the paper is that it does not directly assess whether particle position really correlates with protein denaturation, orientation and reconstruction quality.

Essential revisions:

The authors suggest that the preferred orientation of particles in many cases is due to the exposure at the air-water interface. To this end it would be very helpful if they demonstrate through sub-tomogram averaging, e.g. as they show in Figure 8, that indeed the particles at the interface assume more preferred orientations compared to particles in the middle of the ice layer.

The authors state, e.g. in reference to Figure 5, that it is not clear whether observed protein fragments are from denaturation at the air-water interface, unclean preparation conditions, or protein degradation in solution. This is an important point. Can the authors show that in the majority of these cases there was no degradation in solution, e.g. by SDS PAGE or even negative stain visualization? Could the authors design an experiment showing that particles away from the air water interface are better than those at the air water interface?

It is disappointing that there is no overall analysis of whether sample distribution has an effect on the final 3D reconstruction. Could this be done?

The authors discuss that "protein network films may not be particle friendly", and suggest this might contribute to artefactual reconstructions. This is all rather speculative, but could this be assessed? There must be 3D reconstructions of most of such samples, is an effect seen in the 3D reconstructions where a protein network film is present?

The authors discuss that "particle adsorption implies preferred orientation". There is no explanation as to how preferred orientation was assessed or quantified. Could the authors design an experiment to show whether preferred orientation does correlate with interaction with the air water interface?

The discussion on collection and processing limits (illustrated in Figure 6) could also be followed up in the 3D structures – where the sample is tilted or thick, then changing away from whole-image based CTF estimation toward tilted or per-particle CTF should improve the reconstruction?

Some of the sample properties that the authors suggest should be measured by cryoET can typically also be assessed directly from 2D images. For example, one paragraph is spent on the observation that some areas have overlapping particles. This can be often assessed in 2D for the majority of samples. Concerning the statement "without previously characterizing the sample in the grid holes by cryoET, collection in these areas might severely limit the number of alignable particles due to projection overlap" – many users would see the overlapping particles in 2D images and decide to collect elsewhere. Similarly, people have used CTFtilt in the past to assess whether the field of view is tilted relative to the beam, and estimate the gradient. Tilted exposure areas are not a serious problem for all processing pipelines. Collecting some tomograms can provide a quick and relatively easy sample assessment, and it is a good idea to do this, it is not the case that users who do not do this are doomed to collecting bad images. Please provide a discussion along these lines.

The claims in the subsection “A significant fraction of areas in holes have overlapping particles in the electron beam direction”, about the effects of overlapping particles on cryoEM reconstructions should have citations for each claim or should be left out unless the authors wish to present additional evidence to support the claims.

Technical comments:

In reporting the tilt angles of the specimen in the holes, the authors have provided the magnitude but not the direction of the tilt angles. Even though the tilt direction w.r.t. the grid was unfortunately not tracked, the direction of tilt with respect to the beam axis is important for interpreting the results and statistics so the authors should include the tilt direction for all the tilt angles reported.

If 3D reconstructions of any of the specimen or sub-tomogram particles were performed, plots of the orientation distributions, especially from different regions within the same specimen corresponding to the tomograms presented, should be included. In addition, a more specific reporting of the extent of preferred orientation than "some" "yes" or "unknown" (Table 1) would be helpful.

A detailed description of the measurement of ice thickness, and the determination of the error in this measurement should be reported, i.e. how the value of 10 nm on P14L26 was determined and what are the possible sources of systematic error in these measurements.

For the errors reported in Figure 3, it is not clear if the standard deviation includes the propagated error from the individual measurements or is just the statistical error. This should be described.

For an individual tomogram, the error in the tilt measurement should be determined and included when reporting statistics on multiple tomograms. For example, Figure 3 column 4, the error in an individual tilt angle may be comparable to the value of the angle itself, but one cannot tell currently if this is the case or not.

There is little exploration of the relationships of the parameters that have been measured. For example, is there any correlation between particle size and ice thickness, or between ice thickness and grid type? Given the heterogeneity of the samples, is it appropriate to show average properties (Figure 3), but not explore relationships between parameters?

Comments on presentation:

The manuscript text needs rewriting for readability, at the moment it is very challenging. There are lots of very long sentences and very long paragraphs. The heavy use of "codes" (B2, 2+, M-preferred) is extremely irritating and often unnecessary, forcing the reader to reference figures and tables to understand the text.

The results are all contained within two tables are very difficult to use and understand. Why put three values in one column in brackets running over two lines rather than just three columns or sub-columns? Why use rather cryptic nomenclature like "3+" which requires some thought by the reader to imaging what it might really mean. Entries like "A2, B2 or B3 and B4 and B5‡ (50%), 0°" are hard to interpret, requiring the reader to repeatedly look at legends and other figures to understand anything. The presentation of the data should be rethought.

There are apparent inconsistencies in the manuscript. Some examples: Samples 25 and 46 are said to be ideal, but looking at the schematics in Figure 4, these are not the ideal samples? It seems to me that B1 "Free floating particles (no preferred orientations)" in Table 2 should correlate with + "indicates that there are free-floating proteins" in Table 1 but this does not seem to be the case? How can B3 and B4 be distinguished?

The tables are too small to read when printed – please make fonts larger or put in landscape mode.

The authors often use the term "freely-floating" when they actually mean "randomly oriented" or "randomly positioned." The particles are always under the influence of a variety of forces in solution, some in equilibrium and some not, and are never "freely floating" in liquid water; after vitrification they are stationary.

Throughout the text and tables there is unnecessary complication. Why refer to "ice behaviour (bottom)" as C2, rather than "flat". Why define "N-preferred" and "M-preferred" orientations?

[Editors' note: further revisions were requested prior to acceptance, as described below.]

Thank you for resubmitting your work entitled "Routine Single Particle CryoEM Sample and Grid Characterization by Tomography" for further consideration at *eLife*. Your revised article has been favorably evaluated by John Kuriyan (Senior Editor), a Reviewing Editor, and three reviewers.

We thank the authors for addressing many of our concerns. In particular, the new Figure 5 is very informative and a nice addition. However, it is disappointing that there is no further quantification as regards the effect of the surface-air water interface on the particles and the orientation of the particles. Comments expressed by the reviewers are given below, but note that we are not asking for any further work on the manuscript, except for dataset deposition (see below).

Despite the mixed opinions expressed by reviewers, after discussion among the reviewers and the reviewing editor we have concluded that your paper represents a significant contribution that we will be happy to publish in *eLife*. We are prepared to accept this manuscript as a Tools and Resources article, provided that some of the raw data are publicly deposited. Specifically, we request that both the 2D single particle data and the tilt-series/tomograms be deposited for at least a few specimen, such as, apoferritin, proteasome and maybe another one of the dozens in the tables that are of structures that are already available in the literature. We believe that these deposited datasets will become a resource to the community. When resubmitting the manuscript, please specify which datasets are deposited, and what the accession codes or preliminary processing information is for those datasets.

Detailed comments by the reviewers:

*Reviewer #1:*

I still feel that the authors have done a significant service to the field of single-particle electron cryomicroscopy by presenting this systematic and comprehensive study of single-particle specimen by tomography. It takes several somewhat abstract concepts, like preferential orientation and the air water interface and makes them strikingly obvious and easy to both quantify, and understand for new-comers to the field. The authors have addressed many of the statistical rigour concerns and certainly improved the paper from its previous version. Still, I am somewhat disappointed that the authors have not made better use of this data to address the questions which are not already clear to experts in the field: namely in what percentage of particles, for a particular specimen, and among different specimen, does adsorption to the air-water interface actually prevent structure determination by either destruction or preferential orientation. By quantifying these two outstanding problems the authors could have set a goal and a benchmark for the field as to the current state of the art as well as point toward the opportunity for improvement by some technological solution. Instead we only have the new Figure 5 which sort of hints that it is a mixture, but with no quantification of either effect.

Ultimately, this paper contains important and useful experiments and analysis that should definitely be published. But to the expert reader, the paper sometimes reads a bit like a straw man argument in that the authors claim that most people believe one thing, which is not universally true historically, given section 6 of Dubochet et al. 1988, so their claims about the "wide-ranging implications" of these observations ring less true. This is what makes the lack of quantification of the both the denaturation and orientation distribution of the particles disappointing as it strikes me as a slightly missed opportunity to push the understanding of the problem, and thus the entire field beyond what Dubochet clearly realised more than 30 years ago.

I think the paper can and should be published in *eLife* with minor revision if the authors are willing and able to deposit all the raw data (both tomograms and subsequent 2D micrographs/movies) for at least the specimen shown in the figures. In this way others will also be able to perform analyses to help understand the relationship, if there is one, between the location of a particle in a vitrified specimen and its usefulness in structure determination by cryoEM.

*Reviewer #2:*

I find that the authors have done a good job in addressing most reviewer concerns and comments. The new Figure 5 is very informative and a nice addition. Even though not all comments or suggestions could be addressed with updated experiments, the work is important to the cryo-EM field and the revised manuscript should be published without further delay.

*Reviewer #3:*

The authors have declined to address most of the essential revisions requested by the reviewers, reasonably arguing that the work required is substantial and it would be better to get the paper out there.

Here is the background to my opinion:

There are already some examples of particles clustering at the air water interface from tomograms – the ribosomes in Bharat and Scheres, Nature Protocols 2016, Figure 6 are a good example. It has been assumed since Dubochet, though has not been demonstrated, that particles at the air-water interface may be subjected to denaturation. It has also been broadly accepted in the field that particles showing preferred orientations often due so due to interactions at the air water interface:

"Occasionally particles adopt preferred orientations, presumably due to interactions with the air/water interface.": A primer to single particle cryo-electron microscopy, Cheng et al., 2015.

"In situations where the target adopts a preferred orientation due to interactions with the air-water interface or substrate, there may be a problem similar to the missing wedge problem": Single particle analysis at high-resolution, Cong and Ludtke., 2010.

This manuscript describes an interesting study, showing that the majority of particles in large number samples are located at the air water interface, but it doesn't tell us how detrimental this is for reconstruction, or why it does/doesn't happen. Many of the structures solved from these samples presumably went to high resolution, so this is an interesting question. I think that a descriptive study of the distribution of particles in ice is informative and interesting for the cryo-EM field, but in the first round of reviews we asked to go beyond this and ask how particle position influences the results of the single particle experiment. Without this, I feel that the manuscript is still of interest to the cryo-EM field, but does not have the importance that is typical of papers in *eLife*.

---

## [Author Response]

Summary:Noble et al. applied cryo-electron tomography in a large variety of single-particle samples prepared with a range of vitrification devices and grids to characterize the particle distribution in vitreous ice and the ice geometry in vitrified layers. The results demonstrate convincingly that the vast majority of particles in most cases adsorb to the air-water interface, a phenomenon with potential consequences regarding protein denaturation or complex dissociation, non-physiological conformational changes, as well as preferred particle orientation based on the preferential exposure of the most hydrophobic regions. This study, being the first to systematically examine particle localization in vitrified ice, is important given its implications for the rapidly developing cryo-EM field. However, a weakness of the paper is that it does not directly assess whether particle position really correlates with protein denaturation, orientation and reconstruction quality.

We thank the reviewers and the editors for considering this manuscript, and for their helpful comments and suggestions. We note that the comment on the weakness of the paper above is echoed many times below in various forms. It is clear that the reviewers really want to know whether the tendency of particles to adhere to an air-water interface affects the integrity of the particles and thus the quality of 3D maps derived from these particles. As shown in the manuscript, the vast majority of particles for all samples are absorbed to the air water interface. An experiment to distinguish the quality of maps derived from non-adsorbed vs. adsorbed particle would involve a prohibitive amount of images to accumulate enough non-adsorbed particles and also an enormous amount of additional processing and analysis. We respectfully propose that this new work proposed is not essential to support the major conclusions of our paper (which is that most particles reside at the air-water interface), that it would entail a major amount of work that would significantly delay the publication of this paper, and that it is beyond the scope of the current paper which makes no claims or conclusions regarding whether particle position correlates to protein denaturation or reconstruction quality. Since submitting this paper, we have been doing additional work on some of the issues of interest to the reviewers and some discussion on these matters was presented in a different manuscript now available on bioRxiv (https://doi.org/10.1101/288340). We do intend to try and address whether the quality of maps derived from non-adsorbed particles is different from that of adsorbed particles by using the methods described in that paper which aims to increase the number of non-adsorbed particles and thus reduce the onerousness of this task. We also want to note that it is of course clear that while 90% of particles adhere to the air-water interface, for several of these samples we can readily reconstruct sub 3A maps. Thus, we believe it is very likely that the integrity of particles will be highly sample dependent. But as this is pure speculation and we are not yet able to undertake the necessary experiments to prove this, we prefer not to discuss this further regarding this point.

With that stated we do note that indeed particle position appears to show some correlation with preferred orientation, as can be seen by direct observation of the particles in the tomograms. An analysis of particle position versus orientation and reconstruction was also partially addressed in the bioRxiv manuscript:

https://doi.org/10.1101/288340. This has been cited in the revised manuscript.

These points are addressed further below in response to various specific related comments by the reviewers.

Essential revisions:The authors suggest that the preferred orientation of particles in many cases is due to the exposure at the air-water interface. To this end it would be very helpful if they demonstrate through sub-tomogram averaging, e.g. as they show in Figure 8, that indeed the particles at the interface assume more preferred orientations compared to particles in the middle of the ice layer.

We thank the reviewers for suggesting that the potential change in preferred orientation between particles adsorbed at the air-water interface and those not adsorbed should be shown explicitly. To this end, we have included an additional figure, new Figure 5, with examples of particles, observed directly in the tomograms, showing preferred orientation with respect to the air-water interface when adsorbed but without preferred orientation when not adsorbed (HIV-1 trimer, rabbit muscle aldolase, and T20S proteasome) and without preferred orientation when adsorbed (DnaB helicase-helicase loader). We note that the orientation of particles as observed in tomograms can be determined directly without the need for sub-tomogram processing for most of the samples reported on here. These tomograms are available in the tomogram videos accompanying this manuscript and from the deposited EMDB and EMPIAR datasets. As discussed above, we respectfully suggest that performing sub-tomogram averaging would entail an onerous amount of additional work and substantially delay the publication of this paper. (Note that the references to figures in this response to reviewers are with respect to the figure order in the original submission. The additional figure is referred to as ‘new Figure 5’.)

We have weakened the sub-section name in response to this suggestion. It now reads: ‘Particle adsorption sometimes implies preferred orientation’. We have added references to new Figure 5 in this section.

We note that nearly all of the particles depicted in Figure 8 and the video for Sample #33 in the original manuscript and in the entire dataset (EMPIAR-10135) are adsorbed to air-water interfaces. Virtually no particles in this dataset were in the middle of the ice layer. This has been clarified in the figure caption.

The authors state, e.g. in reference to Figure 5, that it is not clear whether observed protein fragments are from denaturation at the air-water interface, unclean preparation conditions, or protein degradation in solution. This is an important point. Can the authors show that in the majority of these cases there was no degradation in solution, e.g. by SDS PAGE or even negative stain visualization? Could the authors design an experiment showing that particles away from the air water interface are better than those at the air water interface?

We do not have the means to assess the majority of samples reported on in the manuscript as much of the data was gathered from users of our facilities who agreed to have their samples evaluated by tomography during data collection sessions otherwise devoted to single particle analysis.

As shown in this paper, the vast majority of particles for all samples are absorbed to the air-water interface. An experiment to distinguish the quality of maps derived from non-adsorbed vs. adsorbed particle would thus involve a prohibitive amount of additional work and be well beyond the scope of the current paper. We instead intend to address this at a future data by using the methods described in a different paper (https://doi.org/10.1101/288340) that aims to increase the number of non-adsorbed particles.

In response to what we see as the heart of the reviewers’ questions here, we suspect, based on the food and colloidal literature cited in the Introduction, that proteins might undergo minor conformational changes in secondary structure and random coils exposed to the air-water interface (see specifically Table 2 in Yano, Journal of Physics: Condensed Matter, 2012 for a review of observed secondary structural change between bulk proteins in solution and proteins adsorbed to an air-water interface). These higher-resolution structural changes are very unlikely to be resolvable with sub-tomogram processing of the data presented and included in this manuscript. We also note that while the majority of particles like apoferritin and the T20S proteasome are undoubtedly adsorbed to air-water interfaces, we nevertheless routinely produce sub-3Å maps from these structures, so any differences due to the air-water interface are unlikely to be resolved by sub-tomogram averaging.

It is disappointing that there is no overall analysis of whether sample distribution has an effect on the final 3D reconstruction. Could this be done?

This point has been partly addressed above. Also, please see Tan et al., 2017 (cited in our paper) for an explicit example of how conventional untilted single particle cryoEM on a particle, hemagglutinin, with a very high degree of preferred orientation (due to air-water interface adsorption, as shown in this manuscript) affects the final 3D reconstruction compared to tilted collection.

In the bioRxiv preprint (https://doi.org/10.1101/288340) discussed above, we showed that decreasing the time from spot to plunge affects particle orientation at air-water interfaces with regards to the final 3D reconstruction. In that manuscript, we also show how plunging faster can increase the number of non-adsorbed particles for some samples.

The authors discuss that "protein network films may not be particle friendly", and suggest this might contribute to artefactual reconstructions. This is all rather speculative, but could this be assessed? There must be 3D reconstructions of most of such samples, is an effect seen in the 3D reconstructions where a protein network film is present?

We thank the reviewers for allowing us to clarify this point. While we acknowledge that this is speculation in terms of its effects on cryoEM maps, and indeed state this in the manuscript, we also cite several sources in food and colloidal studies that show that proteins denature at air-water interfaces, potentially causing conformational changes. For instance, Meinders, Bosch and Jongh, 2001 shows that by increasing bulk protein concentration in solution, a protein film of increasing thickness forms on the air-water interface consisting of ‘a single homogeneous protein solution.’ This implies that subsequent proteins adsorbing to the already existing protein film at the air-water interface denature. Also, Table 2 in Yano, 2012 cites several results in the literature showing that protein secondary structure at air-water interfaces has been observed to transmute between beta sheets, alpha helices, and random coils.

We thank the reviewers for suggesting that the protein network film might be visible in 3D reconstructions of particles adsorbed to the air-water interface with preferred orientation. We examined the 3D reconstruction of a severely preferentially oriented particle, hemagglutinin, to see if there is a visible protein network. The 3D reconstruction of hemagglutinin was collected at tilts to fill in Fourier space (EMD-8731), as described in Tan et al., 2017. Increasing the threshold of the map in Chimera does not appear to reveal any coherent protein network film. We have not included this assessment in the manuscript because it is inconclusive.

The authors discuss that "particle adsorption implies preferred orientation". There is no explanation as to how preferred orientation was assessed or quantified. Could the authors design an experiment to show whether preferred orientation does correlate with interaction with the air water interface?

This has been partially addressed in the previous discussion above. Preferred orientation was assessed by direct examination of the tomograms (deposited in EMDB and EMPIAR) containing particles whose orientations can be determined. The preferred orientations observed in these tomograms are reflected in the cross-sectional depictions accompanying each video and in Figure 4. We have added a sentence to the ‘Cross-sectional depictions’ section to clarify this for the reader.

Regarding the correlation between air-water interface interaction and preferred orientation, several tomograms in this manuscript where particle orientation can be determined directly explicitly show this correlation. For instance, the tomogram of Sample #5, HIV-1 trimer, shows that the vast majority of the trimers are adsorbed to the air-water interface with the 3-fold symmetric view facing the interface, while the few particles away from the interfaces are in random orientations. The tomogram video of Sample #42, T20S proteasome, show that the vast majority of particles are adsorbed to the air-water interface in the proteasome’s side-view, while the few non-adsorbed particles are in random orientations. We have added a figure, new Figure 5, showing the preferred orientation variation by location of a selection of examples, including the HIV trimer and proteasome. We have also added a reference in the conclusion to our recent bioRxiv preprint (https://doi.org/10.1101/288340), which shows explicit examples of how air-water interface interaction affects preferred orientation.

The discussion on collection and processing limits (illustrated in Figure 6) could also be followed up in the 3D structures – where the sample is tilted or thick, then changing away from whole-image based CTF estimation toward tilted or per-particle CTF should improve the reconstruction?

We understand that the reviewers are requesting that we illustrate that using per-particle CTF processing results in better 3D structures.

Several times throughout the manuscript, we refer to Tan et al., 2017, which explicitly details how per-particle CTF estimation and correction is essential for processing tilted images.

We have added a reference to Mingxu Hu’s upcoming software, Thunder, to the end of the section titled “Most air-water interfaces are tilted with respect to the electron beam” (Hu et al. 2018 https://www.biorxiv.org/content/early/2018/05/23/329169). This software refines per-particle CTF based on a maximum likelihood model. The authors show that their per-particle CTF processing on the public EMPIAR dataset of 2.8 Å T20S proteasome by Campbell et al., 2015 results in a 2.35 Å structure. This T20S sample and the T20S shown in this manuscript were both sourced from the lab of Yifan Cheng. We have also added references in the same location to existing per-particle CTF estimation and correction software, gCTF and CisTEM.

Some of the sample properties that the authors suggest should be measured by cryoET can typically also be assessed directly from 2D images. For example, one paragraph is spent on the observation that some areas have overlapping particles. This can be often assessed in 2D for the majority of samples. Concerning the statement "without previously characterizing the sample in the grid holes by cryoET, collection in these areas might severely limit the number of alignable particles due to projection overlap" – many users would see the overlapping particles in 2D images and decide to collect elsewhere. Similarly, people have used CTFtilt in the past to assess whether the field of view is tilted relative to the beam, and estimate the gradient. Tilted exposure areas are not a serious problem for all processing pipelines. Collecting some tomograms can provide a quick and relatively easy sample assessment, and it is a good idea to do this, it is not the case that users who do not do this are doomed to collecting bad images. Please provide a discussion along these lines.

Thank you for pointing out that we are over-emphasizing the dangers here. However, from our own experience, we cannot agree with the statement that overlapping particles can always be assessed directly from 2D images, particularly for novel samples where the shape of the particle is unknown, for small particles, and for very crowded fields of view. For example, rabbit aldolase, Sample #21 (1 layer) and #22 (2+ layers), are difficult or impossible to differentiate in the 2D projection images.

With regards to somewhat tilted exposures areas significantly degrading single particle collection quality, we did not state this in in the manuscript and did not intend to make this point. We state “In most cases, these tilts are not systematic with respect to particle orientation in the ice, and thus contribute beneficially to angular particle coverage.” We also suggest that collecting with a tilt can be beneficial, e.g. in Figure 7 discussion and in the several references to Tan et al., 2017.

The claims in the subsection “A significant fraction of areas in holes have overlapping particles in the electron beam direction”, about the effects of overlapping particles on cryoEM reconstructions should have citations for each claim or should be left out unless the authors wish to present additional evidence to support the claims.

We have softened the statement to say “may” rather than “will” cause issues. We do not see how this should be cited (do the reviewers have any suggestions?) as these are simple statements about the geometry and the method of projection matching.

Technical comments:In reporting the tilt angles of the specimen in the holes, the authors have provided the magnitude but not the direction of the tilt angles. Even though the tilt direction w.r.t. the grid was unfortunately not tracked, the direction of tilt with respect to the beam axis is important for interpreting the results and statistics so the authors should include the tilt direction for all the tilt angles reported.

Every dataset released on EMPIAR includes the aligned tilt-series of each tomogram reported on with regards to tilt angle with respect to the beam. AppionProtomo estimates and records the three Euler angles required to orient the normal of the specimen to the normal of the Protomo WBP reconstruction. The plots of each of these angles are included in the protomo_alignments/tiltseries####/media/angle_refinement/series####_orientati on.gif images on EMPIAR. Psi is the in-plane rotation required to reach the tilt direction, theta is the tilt with respect to the electron beam, and phi is in-plane rotation required to return to the tomogram coordinate system. The directional angle, Psi, appears to have no correlation with microscope: for example, EMPIAR-10138 (collected on Krios#01 at NYSBC), tiltseries0002 is a hole-magnification tomogram collected near the edge of the grid with (psi, theta, phi) ~= (100, -2, -100), while tiltseries0008 is a high magnification tomogram collected near the center of the grid with (psi, theta, phi) ~= (-50, 10, 50). As a second example on the same Krios microscope, tiltseries0004 and 0005 of EMPIAR10143 were each collected near the center of the grid and have (psi, theta, phi) of about (-40, -1, 40) and (-25, -6, 25), respectively. From the tilt-series we have checked for the microscopes used in this manuscript, the tilt magnitudes and directions do not appear to correlate with the microscope they were collected on; i.e. they are not systematic with respect to the microscope. The main cause of tilt area direction and magnitude appears to be due to variations in local bends in the grid and local ice curvature, and thus particle layer curvature. We have added a sentence stating that “there is no apparent correlation between microscope and tilt direction or magnitude” (section ‘Analysis of single particle tomograms’) and have pointed interested readers to the tomogram tilt orientation plots of all deposited data in the ‘Data deposition and software availability’ section.

If 3D reconstructions of any of the specimen or sub-tomogram particles were performed, plots of the orientation distributions, especially from different regions within the same specimen corresponding to the tomograms presented, should be included. In addition, a more specific reporting of the extent of preferred orientation than "some" "yes" or "unknown" (Table 1) would be helpful.

Again, this has been partly addressed above. Apart from the DnaB helicase-helicase loader tomograms, no other sub-tomogram processing was performed. We have only a few single particle Euler angle distributions and we do not feel these would add substantially to the discussion as these seem to be highly sample dependent (e.g. hemagglutinin is completely preferred while for the T20S the symmetry presumably compensates for the preferred orientation). Again, we feel that this request is well beyond the scope of this paper. We again note that the provided tomogram videos and tomograms on EMDB and EMPIAR show explicit orientations of many of the samples by going slice-by-slice through the tomograms. We have added a sentence clarifying this in the ‘Analysis of single particle tomograms’ section: “For many of the samples shown here and made available in the data depositions, particle orientations can be explicitly determined by direct visualization.”

A detailed description of the measurement of ice thickness, and the determination of the error in this measurement should be reported, i.e. how the value of 10 nm on P14L26 was determined and what are the possible sources of systematic error in these measurements.

We thank the reviewers for prompting us to include proper descriptions of estimations and errors. A section in the Materials and methods titled ‘Estimations and measurement error’ has been added.

For the errors reported in Figure 3, it is not clear if the standard deviation includes the propagated error from the individual measurements or is just the statistical error. This should be described.

A section in ‘Estimations and measurement error’ has been added to describe error propagation.

For an individual tomogram, the error in the tilt measurement should be determined and included when reporting statistics on multiple tomograms. For example, Figure 3 column 4, the error in an individual tilt angle may be comparable to the value of the angle itself, but one cannot tell currently if this is the case or not.

The measurement error is about 1 degree for each tomogram measured, which is the estimated error in orienting a particle layer to the viewing area in 3dmod’s ‘Slicer’ window. 1-5 tomograms per sample were measured for the data presented in Figure 3. The measurement error for the particle layer tilts in particular is an order of magnitude less than the value of the angle itself. An explanation of this measurement and its error has been added to the ‘Estimations and measurement error’ section in the Materials and methods.

There is little exploration of the relationships of the parameters that have been measured. For example, is there any correlation between particle size and ice thickness, or between ice thickness and grid type? Given the heterogeneity of the samples, is it appropriate to show average properties (Figure 3), but not explore relationships between parameters?

We thank the reviewers for allowing us to clarify. As stated at the outset in the manuscript, the exposure areas were determined by the user, thus correlations versus ice thickness would not decouple user subjectivity. Also, we think that there are many parameters that may be implicated (e.g. grid type, plasma cleaning method, buffer, time between cleaning and sample application, sample incubation time, sample incubation temperature, sample incubation humidity, blot force (if used), blot time (if used), nanowire count, grid thickness, etc.); Most of these were not tracked and analysis of these relationships would require a full, separate study.

Comments on presentation:The manuscript text needs rewriting for readability, at the moment it is very challenging. There are lots of very long sentences and very long paragraphs. The heavy use of "codes" (B2, 2+, M-preferred) is extremely irritating and often unnecessary, forcing the reader to reference figures and tables to understand the text.

We thank the reviewers for pointing this out. We have edited and split up several long sentences throughout the text. We think that the longest paragraph – the introductory paragraph briefly reviewing food and colloidal air-water interface studies – is necessary to keep in-tact as it is a departure from the rest of the Introduction. Furthermore, we do not wish to shorten this paragraph because it is an in-depth mini-review of these adjacent fields of study that, to the best of our knowledge, have not been analyzed to this extent in the cryoEM field, and because nearly all results mentioned are referred to throughout the manuscript.

We strongly believe that abbreviation of repeating information is absolutely necessary in dense tables and figures due to the amount of information that must be condensed for the reader. Every abbreviation and nomenclature is explained in the tables, figures, and their captions.

The results are all contained within two tables are very difficult to use and understand. Why put three values in one column in brackets running over two lines rather than just three columns or sub-columns? Why use rather cryptic nomenclature like "3+" which requires some thought by the reader to imaging what it might really mean. Entries like "A2, B2 or B3 and B4 and B5‡ (50%), 0°" are hard to interpret, requiring the reader to repeatedly look at legends and other figures to understand anything. The presentation of the data should be rethought.

We thank the reviewers for the suggestion. We have divided up the columns with three values into sub-columns.

There are apparent inconsistencies in the manuscript. Some examples: Samples 25 and 46 are said to be ideal, but looking at the schematics in Figure 4, these are not the ideal samples? It seems to me that B1 "Free floating particles (no preferred orientations)" in Table 2 should correlate with + "indicates that there are free-floating proteins" in Table 1 but this does not seem to be the case? How can B3 and B4 be distinguished?

We thank the reviewers for pointing out these inconsistencies. The subsection “Ideal samples are a rarity” explains why samples #25 and #46 are ideal.

With regards to inconsistencies between B1 and ‘+’, we have re-assessed the data and have simplified the tables by only including B1 and not ‘+’ to indicate the presence of non-adsorbed particles.

Whether B3 and B4 can be practically differentiated is not the point of Figure 2. This figure is an attempt to encapsulate all potential particle and ice behaviors in grid holes. As the figure caption states, this figure is based on a similar attempt by Tayler and Glaeser, 2008. Regarding B3 and B4 in particular, there might be proteins that adsorb to air-water interfaces, but take a long time relative to plunging time to begin denaturing. If those proteins are frozen before denaturation, they may have a different set of preferred orientations (N) than if they are frozen during/after denaturation (M). To clarify this, the following sentence has been added to the Figure 2 caption: “B3 is different from B4 if, for example, a particle prone to denaturation is frozen before or after denaturation has begun, thus potentially changing the set of preferred orientations.”

The tables are too small to read when printed – please make fonts larger or put in landscape mode.

The tables have been put in landscape mode. We cannot make the font size any bigger as the tables have many columns. High-resolution tables were uploaded to the *eLife* website during manuscript submission.

The authors often use the term "freely-floating" when they actually mean "randomly oriented" or "randomly positioned." The particles are always under the influence of a variety of forces in solution, some in equilibrium and some not, and are never "freely floating" in liquid water; after vitrification they are stationary.

We thank the reviewers for bringing this inconsistency to our attention. The reviewers are correct that ‘freely-floating’ was used incorrectly as a synonym for ‘randomly oriented/positioned’, when in fact such particles may not be in equilibrium. We have opted to replace instances of ‘free/freely floating’ with ‘non-adsorbed’ or ‘not adsorbed to’.

Throughout the text and tables there is unnecessary complication. Why refer to "ice behaviour (bottom)" as C2, rather than "flat". Why define "N-preferred" and "M-preferred" orientations?

We understand the reviewers concern but we believe that unless we use abbreviations the entire paper will be unnecessarily wordy. As mentioned previously, Figure 2 is intended to show all potential particle and ice behaviors in grid holes. There might be proteins that adsorb to air-water interfaces, but take a long time relative to plunging time to begin denaturing. If those proteins are frozen before denaturation, they may have a different set of preferred orientations (N) than if they are frozen during/after denaturation (M). To clarify

this, the following sentence has been added to the Figure 2 caption: “B3 is different from B4 if, for example, a particle prone to denaturation is frozen before or after denaturation has begun, thus potentially changing the set of preferred orientations.”

[Editors' note: further revisions were requested prior to acceptance, as described below.]

We thank the authors for addressing many of our concerns. In particular, the new Figure 5 is very informative and a nice addition. However, it is disappointing that there is no further quantification as regards the effect of the surface-air water interface on the particles and the orientation of the particles. Comments expressed by the reviewers are given below, but note that we are not asking for any further work on the manuscript, except for dataset deposition (see below).Despite the mixed opinions expressed by reviewers, after discussion among the reviewers and the reviewing editor we have concluded that your paper represents a significant contribution that we will be happy to publish in eLife. We are prepared to accept this manuscript as a Tools and Resources article, provided that some of the raw data are publicly deposited. Specifically, we request that both the 2D single particle data and the tilt-series/tomograms be deposited for at least a few specimen, such as, apoferritin, proteasome and maybe another one of the dozens in the tables that are of structures that are already available in the literature. We believe that these deposited datasets will become a resource to the community. When resubmitting the manuscript, please specify which datasets are deposited, and what the accession codes or preliminary processing information is for those datasets.

Two single particle datasets have been added to EMPIAR as the reviewers requested and the manuscript has been edited to reflect the additions.